# Quantifying errors in surface ozone predictions associated with clouds over CONUS: A WRF-Chem modeling study using satellite cloud retrievals

**Young-Hee Ryu[1], Alma Hodzic[1,2,*], Jerome Barre[1,a], Gael Descombes[1], Patrick Minnis[3]**

[1]*National Center for Atmospheric Research, Boulder, CO, USA*

[2]*Laboratoire d'Aérologie, Observatoire Midi-Pyrénées, CNRS, Toulouse, France.*

[3]*NASA Langley Research Center, Hampton, VA, USA*

[a]*now at: European Centre for Medium-Range Weather Forecasts, Reading, United Kingdom*

[1,*]*Correspondence to A. Hodzic:* ***alma@ucar.edu***

*Key words: surface ozone, photolysis, satellite clouds, WRF-Chem*

18                                                    **Abstract**

Clouds play a key role in radiation and hence $O_3$ photochemistry by modulating photolysis rates
and light-dependent emissions of biogenic volatile organic compounds (BVOCs). It is not well
known, however, how much error in $O_3$ predictions can be directly attributed to error in cloud
predictions. This study applies the Weather Research and Forecasting with Chemistry (WRF-
Chem) at 12 km horizontal resolution with the Morrison microphysics and Grell 3D cumulus
parameterization to quantify uncertainties in summertime surface $O_3$ predictions associated with
the cloudiness over contiguous United States (CONUS). All model simulations are driven by
reanalysis of atmospheric data and reinitialized every 2 days. In sensitivity simulations, cloud
fields used for photochemistry are corrected based on satellite cloud retrievals. The results show
that WRF-Chem predicts about 55% of clouds in the right locations and generally underpredicts
cloud optical depths. These errors in cloud predictions can lead up to 60 ppb overestimation in
hourly surface $O_3$ concentrations on some days. The average difference in summertime surface
$O_3$ concentrations derived from the modeled clouds and satellite clouds ranges from 1 to 5 ppb
for maximum daily 8-h average $O_3$ (MDA8 $O_3$) over CONUS. This represents up to ~40% of the
total MDA8 $O_3$ bias under cloudy conditions in the tested model version. Surface $O_3$
concentrations are sensitive to cloud errors mainly through the calculation of photolysis rates (for
~80%), and to a lesser extent to light-dependent BVOC emissions. The sensitivity of surface $O_3$
concentrations to satellite-based cloud corrections is about 2 times larger in VOC-limited than
$NO_X$-limited regimes. Our results suggest that the benefits of accurate predictions of cloudiness
would be significant in VOC-limited regions which are typical of urban areas.

# 1. Introduction

Ozone ($O_3$) is a secondary pollutant that is formed by chemical reactions involving nitrogen oxides ($NO_X = NO + NO_2$) and volatile organic compounds (VOCs) in the presence of ultraviolet radiation. Because $O_3$ is a harmful pollutant and a greenhouse gas, there have been numerous efforts aimed at improving $O_3$ predictions in air quality models, i.e. through a better characterization of the emissions of $O_3$ precursors (Brioude et al., 2013), more detailed chemical mechanisms (Carter, 2010; Sarwar et al., 2013), more realistic lateral boundary conditions (e.g., Tang et al., 2009), and improved representation of meteorological fields with ensemble modeling techniques (Bei et al., 2010; Zhang et al., 2007). A comprehensive review of the current status and challenges of air quality forecasting is given by Zhang et al. (2012). A large $O_3$ bias that still persists in most regional and global models is one of the challenges (Brown-Steiner et al., 2015; Fiore et al., 2009; Im et al., 2015; Lin et al., 2017; Travis et al., 2016). The recent multi-model intercomparison study by Im et al. (2015) indicates that over North America models tend to overestimate hourly surface $O_3$ below 30 ppb by 15–25% and to underestimate $O_3$ levels above 60 ppb by up to ~80%. It is not quantitatively understood how much the individual processes contribute to $O_3$ biases. Among meteorological parameters, clouds can be one of the key factors because they greatly modulate the ultraviolet radiation that is critical for $O_3$ formation. However, they remain one of the largest sources of uncertainties in air quality modeling as Dabberdt et al. (2004) pointed out a decade ago. Accurate cloud predictions in numerical weather models are still challenging, and it has not yet been quantified how much errors in cloud prediction impact surface $O_3$ predictions.

As satellite cloud products have emerged, providing reasonably accurate data with wide coverage and high temporal resolutions in near-real time (e.g., Minnis et al., 2008), they have

been employed in various studies to quantify the effects of clouds on actinic fluxes and/or
photolysis rates (Mayer et al., 1998; Ryu et al., 2017; Thiel et al., 2008). Clouds can greatly
reduce or enhance actinic flux below, above, and inside clouds, and these effects depend mainly
on the cloud optical properties. Ryu et al. (2017) used satellite cloud retrievals of cloud bottom
and top heights and cloud optical depth (COD) in a radiative transfer model, and showed that one
can obtain fairly good (within ±10%) vertical distributions of cloudy-sky actinic flux using
satellite cloud properties. There are, however, only a limited number of studies that have
examined the impact of satellite-constrained clouds and photolysis rates on $O_3$ formation. Pour-
Biazar et al. (2007) and Tang et al. (2015) used satellite-observed clouds to correct photolysis
rates in a three-dimensional chemistry transport model and reported considerable improvement
in surface $O_3$ simulations. Pour-Biazar et al. (2007) showed that the difference in $O_3$ due to the
errors in cloud predictions can be up to 60 ppb for a given pollution episode over the south US.
Tang et al. (2015) showed that 1-month averages of 8-h surface $O_3$ can differ by 2–3 ppb
between the simulations using satellite-derived clouds and model-predicted clouds over the south
US. These studies were performed for rather short time periods (a week or a month) over limited
areas, and provide motivation for a more systematic/comprehensive quantification of the
importance of cloud errors in $O_3$ predictions in summertime and for various chemical regimes.
In the present study, we use satellite-derived COD and cloud boundaries to constrain radiation
fields that impact photochemistry, i.e., photolysis rates and light-dependent BVOC emissions, in
a three-dimensional chemistry transport model (WRF-Chem). Our study targets the contiguous
United States (CONUS) and numerical simulations are performed for June–September 2013. The
WRF-simulated clouds are first evaluated against the Geostationary Operational Environmental
Satellite (GOES) data (section 3). The vertical profiles of $NO_2$ photolysis rates are evaluated
against in-situ airborne measurements during two field campaigns (section 4). The $O_3$ biases
arising from inaccurate cloud predictions are quantified, and discussed in light of the sensitivity
of $O_3$ chemistry to COD (section 5). Unlike the previously mentioned studies, here we quantify
separately the contributions of errors arising from changes in photolysis rates altered by clouds
vs. those arising from light-dependent BVOC emissions to the $O_3$ biases. Conclusions and
discussion are given in section 6.

## 2.  Methodology

### 2.1. Satellite retrievals

The GOES retrievals were performed using the Satellite ClOud and Radiation Property Retrieval
System (SatCORPS), which is an adaptation of the Minnis et al. (2011) algorithms for
application to imagers on all geostationary weather satellites (Minnis et al. 2008) and on NOAA
and MetOp satellites (Minnis et al. 2016). For SatCORPS, the algorithms of Minnis et al. (2011)
were altered as described by Minnis et al. (2010) using the low-cloud height estimation method
of Sun-Mack et al. (2014) and the severely roughened hexagonal column optical model of Yang
et al. (2008) for ice cloud COD retrievals. This study uses a subset of the hourly, 8-km
SatCORPS cloud retrievals from GOES 13 (GOES-East) and GOES 15 (GOES-West) for the
North American domain. The 8-km resolution is achieved by analyzing only every other 4-km
pixel and line. Each pixel is considered to be either 100% cloudy or 100% clear. Of the variety of
cloud properties available, this study only uses cloud bottom height, cloud top height, and COD.
Uncertainties in the cloud products are summarized by Ryu et al. (2017).
Images from coincident times were unavailable for the two satellites: the GOES 13 and GOES
15 data are offset by 15 min. The GOES 13 data taken at UTC + 45 min at every hour were
matched with the GOES 15 data at UTC + 00 min. The pixel-level retrievals were re-gridded to a
12-km resolution to match the WRF-Chem domain (see section 2.2) using the Earth System
Modeling Framework (ESMF) software and the nearest-neighbor interpolation. Because of the
coverage difference between the two satellites, the data of the nearest time from the two satellites
(e.g., 1845 UTC from GOES 13 and 1900 UTC from GOES 15) are merged at 105°W, which is
equidistant from the two sub-satellite longitudes. Only daytime hours (09–23 UTC and 00–04
UTC) are used here.

**2.2. WRF-Chem model simulations**
The present study employs the WRF-Chem model version 3.6.1. with the updated photolysis
scheme. A single domain is used with a horizontal grid size of 12 km (Fig. 1). The
meteorological initial and boundary conditions are provided by the NCEP FNL (Final)
Operational Global Analysis data with a horizontal resolution of 1°, which are available every 6
hours. The model is initialized at 00 UTC 1 June 2013 and spun-up for the first 10 days in the
control simulation (CNTR simulation). The meteorological fields are re-initialized every 48
hours at 06 UTC of a given day to avoid the growth of model errors, and the model is run for 54
hours. Here, the first 6 hours are allowed for spin-up and discarded in each run. The model
outputs for the period of 12 UTC 11 June 2013 through 12 UTC 1 October 2013 are used for the
analysis. As the goal of the study is to use and evaluate the modeled clouds and their impact on
$O_3$ predictions, nudging is not used. This is different from many previous air quality studies that
nudged the meteorology and evaluated modeled $O_3$ with observations. The physics options used
are the Morrison two-moment scheme (Morrison et al., 2009) for the microphysics, RRTMG
scheme for longwave and shortwave radiation (Iacono et al., 2008), MYNN 2.5 level TKE
scheme for the boundary layer parameterization (Nakanishi and Niino, 2006), MYNN surface
layer scheme, Noah land surface model (Chen and Dudhia, 2001), and Grell 3D ensemble
scheme (Grell and Devenyi, 2002) for cumulus parameterization with radiation feedback. The
initial and boundary conditions for chemical species are obtained from the Model for OZone And
Related chemical Tracers (MOZART) global simulation of trace gases and aerosols. For each 2-
day simulation, the chemical state of the atmosphere at 06 UTC is obtained from that at 06 UTC
of the previous simulation. The MOZART-4 mechanism is used for gas-phase chemistry as
described in Knote et al. (2014), and the Model for Simulating Aerosol Interaction and
Chemistry (MOSAIC) aerosol module with 4 bins is used for the aerosol chemistry.
Anthropogenic gas and aerosol emissions are adopted from the AQMEII project in which the
emissions were projected to 2010 from the NEI 2008 inventory (Campbell et al., 2015). Since
Travis et al. (2016) reported that NEI $NO_X$ emissions are too high, we reduced $NO_X$ emission
from all anthropogenic sources by 40% based on their analysis. Note that the $NO_X$ and PAN
from the lateral boundaries are also reduced by 40% in our study. Biomass burning emissions are
taken from the Fire Inventory from NCAR (FINN) (Wiedinmyer et al., 2011). Model of
Emissions of Gases and Aerosols from Nature (MEGAN) (Guenther et al., 2006) version 2.04 is
used for BVOC emissions. As done in Travis et al. (2016) to better match isoprene flux
observations during the Studies of Emissions and Atmospheric Composition, Clouds and Climate
Coupling by Regional Surveys (SEAC[4]RS) field campaign (Toon et al., 2016), we reduced
MEGAN isoprene emissions by 15% over the southeast US. The photolysis rate calculations
utilize the newly implemented TUV option in the WRF-Chem model (Hodzic et al., 2017 in
preparation). This new TUV option uses the updated cross section and quantum yield data based
on the latest stand-alone TUV model version 5.3, and considers 156 wavelength bins with the
resolutions of 1−5 nm. The COD is calculated based on the parameterization given in Chang et al.
(1987), which uses cloud liquid water and/or ice water contents and effective droplet radius
(assumed to be 10 μm both for liquid and ice droplets). To represent subgrid cloud overlaps, a
simple equation of Briegleb (1992) is used, i.e., the effective $COD = COD_0 \times (\text{cloud fraction})^{1.5}$,
where $COD_0$ is the cloud optical depth that is calculated following Chang et al. (1987), and the
cloud fraction is determined based on the relative humidity in a given grid box. According to
Briegleb (1992), applying a power of 1.5 to the cloud fraction is equivalent to the maximum
random overlap.
In the present study, we performed two sets of simulations that use WRF generated clouds in the
CNTR simulation and the GOES clouds in the GOES simulation. The GOES simulations are
conducted from 06 UTC 11 June 2013 through 12 UTC 1 October 2013. The initial chemistry
conditions in the GOES simulation are adopted from the outputs of the CNTR simulation at 06
UTC 11 June 2013. The satellite cloud retrievals are used only to correct photolysis rate and
photosynthetically active radiation (PAR) calculations (i.e., only within the TUV model in WRF-
Chem). That is, the satellite cloud information is not linked to dynamics, microphysics, and
atmospheric radiation. The value of COD is linearly distributed through vertical grids from the
cloud bottom to the cloud top within the TUV model as done in Ryu et al. (2017). This method is
different from the one used in Pour-Biazar et al. (2007) and Tang et al. (2015) in which cloud
bottom height used in their photolysis rate calculations is estimated from the meteorological
model rather than retrieved from the satellite. The use of model estimates can lead to additional
uncertainties in the case of misplaced model clouds compared to observations.
In the present study, PAR calculated from the TUV model is used for the BVOC emissions in
MEGAN for all simulations. This is different from the PAR conventionally used in MEGAN,
which is simply converted/scaled from the downward shortwave radiation from the atmospheric
radiation scheme. In the CNTR (GOES) simulation, the WRF generated clouds (GOES clouds)
are used for the PAR calculation within the TUV model.
To examine the impact of changes in BVOC emissions on surface $O_3$, another set of sensitivity
simulation (EMIS_BVOC simulation) is performed for 10 days (3–12 July 2013), which uses
WRF-generated clouds for the PAR calculation and BVOC emissions as in the CNTR simulation
but uses the GOES clouds for photolysis rate calculations as in the GOES simulation. The
description of the control and sensitivity simulations is summarized in Table 1.

**2.3. Observational data**
*2.3.1. Aircraft data from field campaigns*
We evaluate the model performance using airborne measurements made during two field
campaigns in 2013, i.e., the NOMADSS (Nitrogen, Oxidants, Mercury and Aerosol Distributions,
Sources and Sinks) and the SEAC[4]RS campaigns. The detailed description of the instrument and
measurement data is given in Ryu et al (2017). The NOMADSS campaign was conducted during
1 June–15 July 2013 mainly over the southeast US. We use 16 flight-day data at 1-min time
intervals for the analysis. Data with solar zenith angles larger than 85° are not used. The fire
plume data are filtered out by excluding the data showing $NO_2$ (> 0.1 ppb) or CO (> 120 ppb)
aloft at 4–7 km level. Based on the GOES cloud data, 68% of flight data are characterized by
clear skies and the remaining data (32%) had clouds in the vertical column where the airplane
was located. The SEAC[4]RS campaign also targeted the southeast US although the airplane
sometimes flew over a larger region including California and Midwestern US. The period used
for the analysis is from 6 August through 23 September 2013, which includes 21 flight days. The
time intervals are also 1-min and the data with large solar zenith angles ($> 85°$) and fire plumes
are filtered out. The fraction of data with clouds is 41% for SEAC[4]RS. It is noteworthy that
SEAC[4]RS measurements include large and thick clouds in some cases as a few of the campaign
goals are to identify the role of deep convection in redistributing pollutants and aerosol-clouds
feedbacks, whereas the clouds during NOMADSS were mostly broken clouds.
*2.3.2.   Ground ozone data*
The United States Environmental Protection Agency (EPA) hourly $O_3$ measurements are used for
the analysis. To examine the sensitivity of $O_3$ to COD in different chemical regimes, the VOC-
and $NO_X$-limited regimes are identified using the ratio of $\Delta O_3/\Delta NO_y$, following Sillman and He
(2002). They reported that the $NO_X$-VOC transition occurs when $\Delta O_3/\Delta NO_y = 4$–6. Thus, an
EPA site is denoted as a VOC-limited ($NO_X$-limited) regime when the ratio is less than 4 (greater
than 6). Examples showing the ratio of $\Delta O_3/\Delta NO_y$ for several sites are given in the
supplementary materials (Fig. S1). Among 1,299 EPA sites, 1,062 are used for the analysis: 24%
of the sites are in the VOC-limited and 76% in $NO_X$-limited regimes. The remaining 237 sites are
not used in the present study because those sites fall into the transitional zone, i.e., $\Delta O_3/\Delta NO_y =$
4–6. Note that modeled $O_3$ and $NO_y$ in the CNTR simulation are used to determine whether an
EPA site is in VOC-limited or $NO_X$-limited regime because $NO_y$ measurements are available for
limited sites.
# 3.  Evaluation of WRF clouds with satellite measurements
The model bias in the cloud spatial coverage is evaluated using a 2×2 contingency table (Table
2), where A and D correspond to hit and correct negative events, respectively, and B and C to
false alarm and miss events, respectively. Here, a threshold of 0.3 in hourly COD is used to
distinguish between clear and cloudy sky as the lowest detection limit of satellite retrieved COD
over land is estimated to 0.25 in Rossow and Schiffer (1999), and the use of 0.3 poses slightly
stricter conditions for cloudiness. The agreement index, which is defined as A+D (WRF predicts
correctly cloudy or clear skies), is 69.7% and the probability of detection (POD) for clouds,
A/(A+C), is 55.6%. It is found that the fraction of errors in missing clouds (C, 19.8%) is larger
than that of predicting clouds that are not present in reality (B, 10.4%). The overall bias,
(A+B)/(A+C), is 0.789 and this means that the WRF underestimates the frequency of cloudy
skies. Figure 1 shows the spatial distribution of each contingency category over the CONUS
averaged over the whole study period. In general, the eastern US shows higher cloud frequencies
than the western US except for parts of the Rocky Mountains and the Pacific Northwest. The
largest agreement index appears in central California where the sky condition is mostly clear (Fig.
1d). In terms of errors, the missing clouds rate has its highest frequency (20−35%) in the
Midwestern and northwestern US, while the highest frequency of false alarm (20–30%) occurs
over the southeast US and the southeastern Texas. The sum of category B and C can be found in
the supplementary material (Fig. S2). It should be noted that the contingency categories are
based on binary results of cloud-free or cloudiness and so they do not provide quantitative
comparison of cloud optical properties, e.g., COD. For example, even though the WRF model
produces clouds in the right locations (category A), the WRF CODs can differ from those
retrieved from satellite data.
Figure 2 evaluates quantitatively COD and vertical extent of clouds between the model and
satellite retrievals. The vertical extent of clouds is classified based on the International Satellite
Cloud Climatology Project (ISCCP) definition (Rossow and Schiffer, 1999), which are as
follows: i) low-level: cloud top height ≤ 3 km, ii) mid-level: 3 km < cloud top height ≤ 6 km, iii)
high-level: cloud bottom height > 6 km, and iv) multi-layered or deep convection: cloud bottom
height ≤ 6 km and cloud top height > 6 km. Even though multiple cloud layers can be resolved in
the WRF model, these kinds of clouds are not resolved in the satellite retrievals used in this study.
Thus, for a fair comparison, the multi-layered clouds in the WRF model are not further resolved
into cloud layers. Note that the liquid/ice water contents from cumulus clouds (parameterized
clouds) are included in the model COD calculations.
The frequency distribution of CODs does not have the same shape in the model and observations.
The WRF model overpredicts by a factor of 2 very thin clouds with COD < 1, whereas the
GOES retrievals show that the most abundant clouds have CODs of 2–5. The majority of
optically very thin clouds from the WRF model correspond to high-level cirrus clouds. This is
consistent with the result of Cintineo et al. (2013), showing that the Morrison microphysics
scheme produces too many upper-level clouds by comparing GOES infrared brightness
temperature with the WRF model. Note that the optically-thin multi-layered clouds very likely
contain cirrus clouds because their top height is greater than 6 km. The WRF model produces
fewer clouds with COD > 1 than observed, and the discrepancy is most apparent for optically
very-thick clouds (COD > 50). As a result, the model COD mean and standard deviation are
smaller than those for the retrievals, which are 8.3 and 12.7, respectively for the WRF model,
and 17.8 and 30.8, respectively for the GOES retrievals.
## 4. Impact of cloud errors on photolysis rates
Figure 3 compares the cloudy-sky averaged vertical profiles of $NO_2$ photolysis rates ($JNO_2$)
predicted by WRF-Chem and measured during the NOMADSS (Fig. 3a) and SEAC[4]RS (Fig. 3d)
campaigns. The histograms of ratio of $JNO_2$ simulated to that observed under cloudy conditions
are also shown for the CNTR and GOES simulations.
For both campaigns, the simulations with satellite clouds (GOES simulations) generally show
better agreement with the observed $JNO_2$ profiles than the CNTR simulations, especially above
the boundary layer (above ~2 km). The histograms of the ratio model-to-observation $JNO_2$ also
show a better performance generally in the GOES simulation than in the CNTR simulation: the
mean of the ratio is closer to 1 in the GOES simulation than in the CNTR simulation for
SEAC[4]RS, the standard deviations are reduced in the GOES simulation compared to those in the
CNTR simulation for both campaigns, the root-mean-square-errors are lowered in the GOES
simulation compared to those in the CNTR simulation, and the correlation coefficients are closer
to 1 in the GOES simulation than in the CNTR simulation. For NOMADSS, the large bias in the
highest ratio bin (> 2) is 24% less in the GOES simulation than in the CNTR simulation. The
reduction of the large bias (bin > 2) in the GOES simulation is more substantial for SEAC[4]RS
and reaches 47%. These reductions are attributed to a better representation of the below-cloud
and inside-cloud conditions when satellite clouds are used (not shown). This is because the
number of data influenced by thick clouds is larger in SEAC[4]RS than in NOMADSS and the
measurements in the presence of those thick clouds were mostly made under below-cloud or
inside-cloud conditions.
## 5. Impact of cloud errors on ground level ozone
### 5.1. An example on 8 July 2013 in Midwestern US
Figure 4 shows an example of how model errors in cloud fields impact $O_3$ predictions. This
example includes thunderstorm systems over the Midwestern US. The CNTR simulation misses
clouds or underpredicts CODs over metropolitan Chicago and the region south of Lake Michigan.
This results in the overprediction of $JNO_2$ by up to 0.54 $min^{-1}$ (~90%) compared to that
computed using GOES clouds. The resulting changes in $O_3$ concentration are regional and the $O_3$
overprediction in the plume originating from the Chicago area is up to 62 ppb (~60% of $O_3$ in the
CNTR simulation). As a result of the cloud corrections, $O_3$ in the GOES simulation agrees better
with observations in those regions (compare Fig. 4d with Fig. 4e and Figs. 4g,h,i). The time
series of $O_3$ at the three sites (marked in Fig. 4f) near Lake Michigan show particularly improved
agreement with observations when satellite clouds are used. The large $O_3$ biases of 20.5 ppb at
11 CST at Chicago, IL, 19.2 ppb at 13 CST at La Porte, IN, and 23.5 ppb at 16 CST at Holland,
MI in the CNTR simulation are reduced to 1.7 ppb, 3.2 ppb, and −0.11 ppb in the GOES
simulation, respectively. It is also apparent that the bias reduction in $O_3$ shifts eastward (from
Chicago, IL to Holland, MI) as the thunderstorm moves eastward during the day. An important
implication of this finding is that errors in cloud predictions can lead to wrong $O_3$ alerts in areas
where model does not predict clouds well. For example, the maximum daily 8-h average $O_3$
(MDA8 $O_3$) concentration is 75.3 ppb at Holland, MI in the CNTR simulation (Fig. 4i) and this
value exceeds the $O_3$ standard (70 ppb for MDA8 $O_3$). However, the MDA8 $O_3$ concentration at
the same location is 63.0 ppb in the GOES simulation and 60.4 ppb in the observation. Therefore,
an $O_3$ action alert would have been issued if the CNTR simulation results are used, which results
in a false alarm. The example shown here emphasizes the important roles of clouds in the Great
Lakes region where large $O_3$ biases have been reported previously in air quality forecasts (e.g.,
Cleary et al., 2015). The correction of clouds both over the lakes and in the upstream regions
(mostly large cities located to the west/southwest of the lakes) significantly reduces the $O_3$ bias.
It is also shown that polluted air masses from the source regions can be advected over the lakes
(not shown). In this case in which precursor levels can be high over the lakes, the presence of
clouds over the lakes can greatly affect $O_3$ formation over the lakes.
In general, the regions exhibiting $O_3$ differences between the two simulations coincide with the
regions where $JNO_2$ values are different. More importantly, large $O_3$ differences are found near
urban areas (e.g., Chicago, IL; downwind area of Kansas City, MO; Omaha, NE and its
downwind area). Even though the difference in COD or $JNO_2$ is significant in central Indiana,
for example, the difference in $O_3$ in the region is relatively small compared to that near Lake
Michigan.

**5.2. Maximum daily 8-h average $O_3$**
Figure 5 shows the maps of MDA8 $O_3$ averaged over the study period for the CNTR simulation
and the difference in MDA8 $O_3$ between the CNTR and GOES simulations. The spatial
distribution of MDA8 $O_3$ in the GOES simulation is similar to that in the CNTR simulation (thus
the GOES spatial average is not shown here), but the $O_3$ levels are considerably different. In Fig.
5b, the Midwestern, eastern, and northwestern US regions show the largest $O_3$ differences, up to
5.8 ppb, with lower $O_3$ levels in the GOES simulation. These regions generally belong to the
contingency category C (Midwestern and northwestern US) or category A (eastern US). On the
other hand, the regions with negative differences, i.e., some places over the south/southeastern
US, coincide with the contingency category B. These differences are expected and can be
interpreted as follows: when the WRF model misses clouds (clear sky in the CNTR simulation,
category C) or underestimates COD (as seen in Fig. 2), surface $O_3$ is overestimated. When the
WRF model generates clouds that are not present in reality (clear sky in the satellite retrievals,
category B), surface $O_3$ is underestimated. It should be noted that not all regions belonging to
category B or C have significant $O_3$ differences. Interestingly, the regions exhibiting significantly
large $O_3$ differences coincide with large urban areas, e.g., Seattle, WA; Los Angeles, CA;
Chicago, IL; Cleveland, OH; Houston, TX; New Orleans, LA; Atlanta, GA; and Miami, FL. The
reasons for this result are explored in section 5.4 and 5.5.
The performance of the GOES simulation is found to be better than that of the CNTR simulation
as compared to observations: for example, under cloudy conditions (COD > 20, see section 5.4
for the criterion), the root-mean-square error of MDA8 $O_3$ in the GOES (CNTR) simulation is
13.2 ppb (16.9 ppb) and the correlation coefficient of MDA8 $O_3$ in the GOES (CNTR)
simulation is 0.5 (0.4).
**5.3. Relative contribution to $O_3$ errors from photolysis rates and BVOC emissions**
It is expected that reduced BVOC emissions (especially isoprene) due to the presence of clouds
can also decrease $O_3$ formation. Figure 6 shows the spatial distributions of relative changes in
PAR and isoprene emission between the EMIS_BVOC and GOES simulations averaged over a
10-day period. Because the WRF model tends to underestimate COD or is not able to reproduce
clouds in Midwestern and western US, PAR and biogenic isoprene emissions are larger in the
EMIS_BVOC simulation than in the GOES simulation. On the other hand, the model
overestimates COD or produces clouds that are not present in reality over the southeast US, so
PAR and biogenic isoprene emissions are lower in the EMIS_BVOC simulation than in the
GOES simulation. The change in PAR (biogenic isoprene emissions) resulting from the
difference in clouds fields between the WRF model and satellite retrievals is up to ±30–40%
(±25%). Figure 6d shows the relative $O_3$ difference between EMIS_BVOC and GOES
simulations to $O_3$ difference between CNTR and GOES simulations (Fig. 6c). It is seen that the
contribution of changes in BVOC emissions is considerable only for some regions and it ranges
from ~10–40%. The average contribution of changes in BVOC emissions over land is ~20%
compared to changes of BVOC emissions plus photolysis rates using GOES satellite clouds. The
contribution of BVOC emissions is larger (~40%) in urban areas over the southeast (specifically
in Charlotte, NC). The difference in $O_3$ in Charlotte, NC resulting from changes in BVOC
emissions is about 1.5 ppb and that from changes in both photolysis rates and BVOC emissions
is about 3.5 ppb. In some regions, such as Midwestern, western Pennsylvania, and central New
York, the effect of BVOC emissions is negligible.

### 365    5.4. Cloud effects on ozone bias in VOC- and $NO_X$ -limited regimes

In this section, we examine the effects of clouds on $O_3$ in VOC-limited and $NO_X$-limited regimes
in order to understand the reasons for a stronger $O_3$ response to cloud corrections in urban areas
than in the remote regions. Figure 7 shows how cloud corrections affect $O_3$ errors in different
regimes. Here, MDA8  $O_3$ is used to compute the model $O_3$ bias (simulation minus observation).
Figures 7a and 7b show the probability density functions of the model $O_3$ bias for the CNTR and
GOES simulations, respectively, at all ground sites experiencing considerably thick (COD > 20)
clouds. In this example, an EPA site is considered under cloudy sky conditions when hourly
COD greater than the chosen threshold (here, 20) is present at the site for at least 4 hours within
the 8-h time window in a given day. The decrease in the $O_3$ bias for VOC-limited regime is
significant, and the difference in median values between the two simulations is 5.2 ppb. The
decrease in $O_3$ bias for $NO_X$-limited regimes (2.7 ppb) is about 2 times smaller than that for
VOC-limited regime. An important result is that the frequency of very large biases (e.g., greater
than 20 ppb) is substantially reduced when cloud fields are corrected, especially for the VOC-
limited regime. This implies that more accurate cloud predictions ultimately improve the
accuracy of $O_3$ alert predictions, especially in polluted urban areas.
Figure 7c shows the change in median values of MDA8 $O_3$ bias for a range of COD thresholds.
We find that the $O_3$ bias increases with increasing cloudiness in the CNTR simulation. As
previously mentioned, the $O_3$ bias is generally larger for VOC-limited regimes than for $NO_X$-
limited regimes. When the radiation fields are corrected with satellite clouds, the model $O_3$ bias
is considerably reduced (but not zero). In addition, the $O_3$ bias in the GOES simulation does not
increase as much as that in the CNTR simulation when cloudiness increases. This implies that
there are other sources of $O_3$ biases in the GOES simulation, which are not likely associated with
cloudiness. The other errors sources can be precursor emissions, mixing/transport, and deposition.
Fig. 7d compares the median values of MDA8 $O_3$ bias between the two simulations (CNTR
minus GOES), and shows that the difference in MDA8 $O_3$ between the two simulations clearly
increases as the COD threshold increases and that the effect of cloud correction is larger in VOC-
limited than in $NO_X$-limited regimes. The reduced $O_3$ bias as a result of cloud corrections ranges
from 1 to 5 ppb depending on CODs and chemistry regimes. This represents up to ~40% of the
total $O_3$ bias under cloudy conditions in the current model version (e.g., 5.2 ppb of 12.6 ppb for
COD threshold of 20 in VOC-limited regimes). Note that the results for the sites in transitional
zone (the slope of $\Delta O_3/\Delta NO_y$ is 4–6) showed that the effects of cloud in the transitional zone are
intermediate; that is, larger than those for $NO_X$-limited regimes but smaller than those for VOC-
limited regimes (not shown).
We performed additional analysis by dividing VOC- and $NO_X$-limited sites into groups that have
similar ranges of peak MDA8 $O_3$ concentration during the period of June–September 2013 (Fig.
S3). All sites are grouped into bins with peak value of MDA8 $O_3$ ranging from larger than 75
ppb, 70–75 ppb, 65–70 ppb, 60–65 ppb, to smaller than 60 ppb. The maximum reduction in $O_3$
bias due to cloud corrections is obtained for the VOC-limited sites with peak MDA8 $O_3$ of 65–70
ppb and reaches ~8 ppb. The maximum reduction for $NO_X$-limited sites, on the other hand, is ~4
ppb and found for the sites with peak MDA8 $O_3$ of 70–75 ppb. Although the degree of the $O_3$
bias reduction varies somewhat among the bins for a given ozone regime, the effects of cloud
correction on $O_3$ bias reduction remain larger in VOC-limited regimes than $NO_X$-limited regimes.
We examine the $O_3$ bias over the southeast US where large overpredictions at the surface have
been reported (e.g., Travis et al. 2016) in the supplementary material. It is found that a
considerable portion of $O_3$ bias is attributable to inaccurate cloud predictions over the southeast
US, but the degree of the effects of clouds is smaller than that over CONUS as a whole (Fig. S4).
The maximum reduction in $O_3$ bias due to inaccurate cloud predictions is 4.5 ppb over the
southeast US and 5.3 ppb over CONUS. Still, large $O_3$ biases of ~11 ppb are present over the
southeast US (compared to those of 6–9 ppb over CONUS) even though the clouds and radiation
fields that are relevant to photochemistry are corrected. This result implies that errors resulting
from other processes exist and are responsible for the surface $O_3$ overpredictions over the
southeast US. More in-depth studies that find and quantify errors are therefore required to better
predict the $O_3$ over the southeast US as well as CONUS.

**5.5. Ozone formation sensitivity to changes in photolysis rates**
The difference in $O_3$ sensitivity to changes in photolysis rates (resulting from the presence of
clouds) in different regimes is determined by calculating $dln(O_3)/dln(JNO_2)$ ratios as in
Kleinman (1991). Table 3 lists those sensitivity coefficients of $O_3$ to $JNO_2$ and shows that $O_3$ is
more sensitive to $JNO_2$ in VOC-limited than in $NO_X$-limited regimes, being 1.69 times larger
under cloudy-sky conditions and by 1.65 times greater under clear-sky conditions. Similar
sensitivities were reported for OH by Berresheim et al. (2003) with the sensitivity of OH to $JO^1D$,
$dln(OH)/dln(JO^1D)$, of 0.8 at high $NO_2$ levels (~10 ppb) and 0.68 at low to moderate $NO_2$ levels
(~1 ppb). The corresponding sensitivities from our study are 1.1 for VOC-limited regimes and
0.66 for $NO_X$-limited regimes under clear-sky conditions. Similar results are also found for the
net chemical production of $O_3$ and OH concentration, revealing stronger responses to changes in
cloudiness in VOC-limited regimes than $NO_X$-limited regimes (Fig. 8). It is interesting to note
that OH and $HO_2$ have local maxima at CODs between 2 and 5. As shown in Ryu et al. (2017),
the enhancement of actinic flux at the surface due to optically thin clouds (CODs < 5) is
considerable for high-level clouds, i.e., cirrus. The local maxima, therefore, likely result from the
fact that the GOES clouds have the largest portion of cirrus for CODs of 2–5 as seen in Fig. 2b.
Figure 8 also shows that the variation (defined by 25 and 75 percentiles) of net chemical
production of $O_3$ with respect to COD is much larger in VOC-limited conditions. This result
suggests that predicting $O_3$ under cloudy conditions is likely more difficult in VOC-limited than
in $NO_X$-limited regimes. It is also noticeable that the $HO_2$ radical concentration remains
relatively high in $NO_X$-limited regimes even under cloudy conditions as compared to the VOC-
limited regimes. Note that the results of WRF-Chem here include the effects of both photolysis
rates and BVOC emissions.
A simplified box model (BOXMOX, Knote et al. (2015)) simulation using the same chemical
mechanism (MOZART-4) as WRF-Chem was performed to better understand $O_3$ sensitivity to
changing cloudiness in different chemistry regimes. The emission rates for VOC-limited ($NO_X$-
limited) regime are those of the Chicago urban (rural) area in the WRF-Chem simulation. The
initial conditions are taken from the CNTR simulation at 09 CST 7 July 2013 in the Chicago
suburban area for both regimes. Dry deposition is not considered. Photolysis rates for all species
that are photodissociable are varied from clear-sky to cloudy conditions with up to 80%
reduction. The 80% reduction roughly corresponds to COD of 35 (not shown). The box model is
integrated for 3 hours and photolysis rates are kept constant during the simulation (i.e., no
diurnal variations). The box model results are found to be consistent with the results from the
WRF-Chem simulations: the variations of $O_3$ and OH with respect to decreasing photolysis rates
are larger in VOC-limited regime than in $NO_X$-limited regime (Fig. S5, in the supplementary
material). Note that the net chemical production of $O_3$ obtained from the box model results also
shows a larger sensitivity to cloudiness in VOC-limited regimes than in $NO_X$-limited regimes,
which is similar to Figs. 8a and 8d (not shown). Figure 9 shows production and loss terms of
$RO_X$ (= OH + $HO_2$ + $RO_2$) radicals with variations in photolysis rates for VOC-limited and $NO_X$-
limited regimes. In both regimes, the decreased sunlight due to clouds reduces OH formation by
photodissociation of $O_3$ (primary source of OH). The larger sensitivity of OH radicals to COD in
VOC-limited regimes as seen in Fig. 8 is associated with the loss of OH by the radical
termination reaction between OH and $NO_2$ under $NO_X$-rich conditions, which leads to the large
decrease in OH (Fig. 9a). On the other hand, in $NO_X$-limited regimes, the radical termination
reactions are the radical-radical reactions (Fig. 9b). In this regime, OH mainly reacts with VOCs
and propagates through radical cycles by producing $HO_2$/$RO_2$ radicals, rather than being
terminated by the reaction with $NO_2$. Given that the reaction between NO and $HO_2$ becomes the
largest source of OH budget (secondary source of OH) at an $NO_X$ concentration of ~1 ppb
(Ehhalt and Rohrer, 2000; Eisele et al., 1997), OH can be relatively less sensitive to the changes
in radiation. Note that the mean daytime $NO_X$ concentration over CONUS in $NO_X$-limited
regimes is 1.2 ppb and that in VOC-limited regimes is 6.7 ppb for this study period. Another
attribute is a greater contribution of $H_2O_2$ photodissociation to the production of $RO_X$ in $NO_X$-
limited regimes than that of $HNO_3$, which is negligible. Unlike the radical terminated in VOC-
limited conditions, a non-negligible amount of terminated radicals can be recycled in the $NO_X$-
limited regime.

## 6. Sensitivity of cloud optical depth and $O_3$ to microphysics and convective schemes

It should be emphasized that our study was performed using a specific representation of the
cloud microphysics by Morrison et al. (2009) and cumulus parameterization (Grell and Devenyi,
2002). To test the robustness of our results with regard to the representation of clouds, another
microphysics scheme, Thompson scheme (Thompson et al., 2008), is employed for a 10-day (3
July–12 July 2013) sensitivity simulation. The COD comparison in Fig. S6 shows that with the
Thompson scheme the model predicts fewer clouds for all ranges of CODs as compared to
GOES retrievals, except for the very thin ones (COD < 1) in which the number of those clouds is
still overpredicted as seen in the simulation with Morrison scheme. Compared to the Morrison
scheme, the Thompson scheme produces significantly less high-level (cirrus) clouds. This is also
consistent with the findings of Cintineo et al. (2013). Despite this difference, the shape of the
COD distribution from the two microphysics schemes are rather similar to each other.
The MDA8 $O_3$ bias with the Thompson scheme is evaluated (Fig. S7), and compared to that of
the Morrison scheme for the same period. Under the conditions of COD greater than 20, for
example, the baseline simulation with the Thompson scheme (that uses model generated clouds)
shows that a median bias (14.79 ppb) is a bit smaller than that with the Morrison scheme (16.22
ppb) for that period in VOC-limited regimes. In the sensitivity simulation with the Thompson
scheme that uses GOES satellite clouds for photochemistry, the median bias is reduced by 5.45
ppb (~37%, Fig. S7a) in VOC-limited regimes and by 2.06 ppb (~20%, Fig. S7c) in $NO_X$-limited
regimes, which are consistent with the results of our base simulation. The degree of the effects of
cloud correction in the sensitivity simulations with the Thompson scheme, ranging from 0.5 to
5.5 ppb, is similar to that found in our base simulation with the Morrison scheme. Therefore, the
general conclusions remain the same: i.e., errors in $O_3$ predictions resulting from errors in cloud
predictions are considerable (up to ~5 ppb on average) and the effects of cloud corrections are
larger in VOC-limited regimes than in $NO_X$-limited regimes.
To estimate the sensitivity of our results to cumulus parameterization schemes, sensitivity
simulations with the Grell-Freitas scheme (Grell and Freitas, 2014) are performed. As done for
microphysics scheme, a period of 10 days (3–12 July 2013) was considered. In Fig. S8, the
histograms of cloud optical depths obtained for the 10-day period from Grell-Freitas scheme and
from Grell-3D scheme show that the distributions of cloud optical depths are in general similar
to each other. The Grell-Freitas scheme tends to produce fewer clouds with small or moderate
cloud optical depths (Fig. S8). As shown in Fig. S9, the degree of cloud correction in reducing
$O_3$ bias is larger in VOC-limited regimes than in $NO_X$-limited regimes in the simulation with
Grell-Freitas scheme, and thus the conclusions originally drawn remain unchanged.

## 7. Conclusions and discussion

We performed quantitative analyses of the WRF-Chem model meso-scale (12 km) simulations to determine how much errors in cloud predictions contribute to errors in surface $O_3$ predictions during summertime over CONUS. Clouds were generated using the Morrison microphysics and Grell 3D cumulus parameterization schemes. It is found that the WRF-Chem model is able to generate roughly 55% of the clouds in the right locations by comparing to satellite clouds. A quantitative comparison of COD shows that the WRF-Chem model predicts too many thin cirrus clouds with CODs less than 1, and also considerably underpredicts the optical depths for a majority of cloud systems.

The errors in cloud predictions can lead to large hourly $O_3$ biases of up to 60 ppb, for example, for specific cases in which the model misses deep convective clouds that are present in reality. On average, the errors in MDA8 $O_3$ of 1–5 ppb are found to be attributable to errors in cloud predictions under cloudy sky conditions. We quantify separately the contribution of changes in photolysis rates and emissions of light-dependent BVOCs to cloud-related errors in surface $O_3$. The contribution of photolysis rates to surface $O_3$ is larger (~80% on average) than that of BVOC emissions. The contribution of BVOC emissions to $O_3$ can become important (~40%) in the VOC-limited regimes where BVOC emissions are large (i.e., cities of the southeast US).

The effects of cloud corrections are more impactful in VOC-limited (or high-$NO_X$) than in $NO_X$-limited (or low-$NO_X$) regimes. The sensitivity of $O_3$ with respect to COD is about 2 times larger in VOC-limited than in $NO_X$-limited regimes. This finding is consistent with the box modeling results that were performed for typical urban/rural conditions under varying photolysis rates. The production of radicals (OH, $HO_2$, and $RO_2$) decreases with decreasing photolysis rates in the presence of clouds. The primary reason for the larger sensitivity of $O_3$ formation to clouds in

VOC-limited regimes is that the loss of OH is much stronger in VOC-limited regimes due to the
reaction with $NO_2$. Thus, OH cannot readily propagate through the radical cycles. In $NO_X$-
limited regimes, the radicals terminated from the radical cycles are mostly $HO_2$ and $RO_2$ rather
than OH. Thus, OH can remain in the cycles and continue to produce $HO_2$ and $RO_2$ by reacting
with VOCs before termination. The interconversion of $HO_2$ to OH is the dominant process in
$NO_X$-limited regimes, and therefore OH and $O_3$ formations are less sensitive to changes in
radiation.
We showed that considerable reduction in $O_3$ bias is achieved by correcting cloud-related
radiation fields; however, $O_3$ is still overpredicted by the WRF-Chem model. The remaining bias
likely results from other processes involved in the $O_3$ lifecycle such as precursor emissions from
both anthropogenic and biogenic sources, transport, turbulent mixing, and dry deposition, which
quantitative assessment is beyond the scope of this study.
One should keep in mind that the quantitative estimate of the $O_3$ bias related to the cloud effects
on radiation as reported in this study could be sensitive to several factors. In particular, this study
is based on a particular configuration of the WRF-Chem model with regard to the radiation,
microphysics, cumulus, boundary layer parameterization and the chemistry scheme. We have
tested the sensitivity of our results to the choice of microphysics and cumulus parameterization
schemes, and have shown that MDA8 $O_3$ biases are reduced by up to ~5 ppb with the satellite
cloud corrections in the simulations with the different microphysics and cumulus
parameterization schemes, which is consistent with the results found in our base simulations.
This study suggests that accurate cloud predictions through data assimilation or cloud mask
corrections with near-real time satellite cloud data would improve the accuracy of $O_3$ predictions
and that the benefit is expected to be greater in VOC-limited than in $NO_X$-limited regimes. It
should be noted that our estimates are based on WRF-Chem simulations that use initial and
boundary conditions from meteorological analysis data, which is an improved estimate of the
meteorological state compared to forecast data, and thus the reduction of errors in $O_3$ predictions
could be even greater in a forecasting setting. From the perspective of $O_3$ forecast, our study
indicates that there is a need for an enhanced understanding of the evolution of errors in $O_3$
forecasts associated with errors in cloud forecasts, and for optimizing the use of meteorological
forecasts to allow more accurate near-term $O_3$ predictions. The present study corrects cloud
fields in WRF using satellite clouds only for radiation that is relevant to photochemistry, and
those cloud corrections do not affect other meteorological variables such as surface temperature,
wind, humidity, boundary layer height, etc. In a future study, we plan to examine the effects of
satellite cloud assimilation on near-term $O_3$ forecasts using enhanced forecasts such as the Rapid
Refresh products from NOAA (Benjamin et al., 2016) that take into account cloud data
assimilation to derive meteorology. The Rapid Refresh uses satellite cloud products as well as
cloud observations from the ground and considers the thermodynamic balance between
temperature and humidity due to the presence of clouds. Thus, this will allow investigating the
effects of cloud assimilation on $O_3$ forecasts not only through changes in radiation for
photochemistry but also through changes in meteorological variables.

**Acknowledgments**
We acknowledge Samuel Hall and Kirk Ullmann for providing actinic flux data that are used for
supplementary analysis, George Grell and Geoff Tyndall for helpful discussions. This study is
supported from NASA-ROSES grant NNX15AE38G. P. Minnis was supported by the NASA
Modeling, Analysis, and Prediction Program. The National Center for Atmospheric Research is
sponsored by the National Science Foundation. We would like to acknowledge high-performance
computing support from Cheyenne (doi:10.5065/D6RX99HX) provided by NCAR's
Computational and Information Systems Laboratory, sponsored by the National Science
Foundation. The GOES cloud retrievals are available at https://satcorps.larc.nasa.gov. The EPA
ozone data can be downloaded at
https://aqsdr1.epa.gov/aqsweb/aqstmp/airdata/download_files.html.

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

Table 1. Description of WRF-Chem simulations.

|  | Photolysis rates | PAR | Analysis Period |
|---|---|---|---|
| CNTR | WRF clouds | WRF clouds | 06 UTC 11 June–12 UTC 1 October |
| GOES | GOES clouds | GOES clouds | 06 UTC 11 June–12 UTC 1 October |
| EMIS_BVOC | GOES clouds | WRF clouds | 06 UTC 3 July–12 UTC 13 July |















Table 2. Contingency table for WRF simulation and GOES satellite clouds. The number of data
for each category is normalized by the total number of data.

| | | GOES Satellite | |
|---|---|---|---|
| | | Cloudy | Clear |
| WRF simulation | Cloudy | A (hit) 24.8% | B (false alarm) 10.4% |
| | Clear | C (miss) 19.8% | D (correct negative) 44.9% |












Table 3. Sensitivity coefficient of $O_3$ to $JNO_2$, i.e., $dln(O_3)/dln(JNO_2)$. The values of
$dln(O_3)/dln(JNO_2)$ for the period of 09–13 LST are averages over only CONUS EPA stations
that have monotonically increasing $O_3$ concentrations with time.

| | Cloudy sky ($5 < COD < 20$) | Clear sky |
| --- | --- | --- |
| VOC-limited | 0.59 | 1.27 |
| $NO_X$-limited | 0.35 | 0.77 |














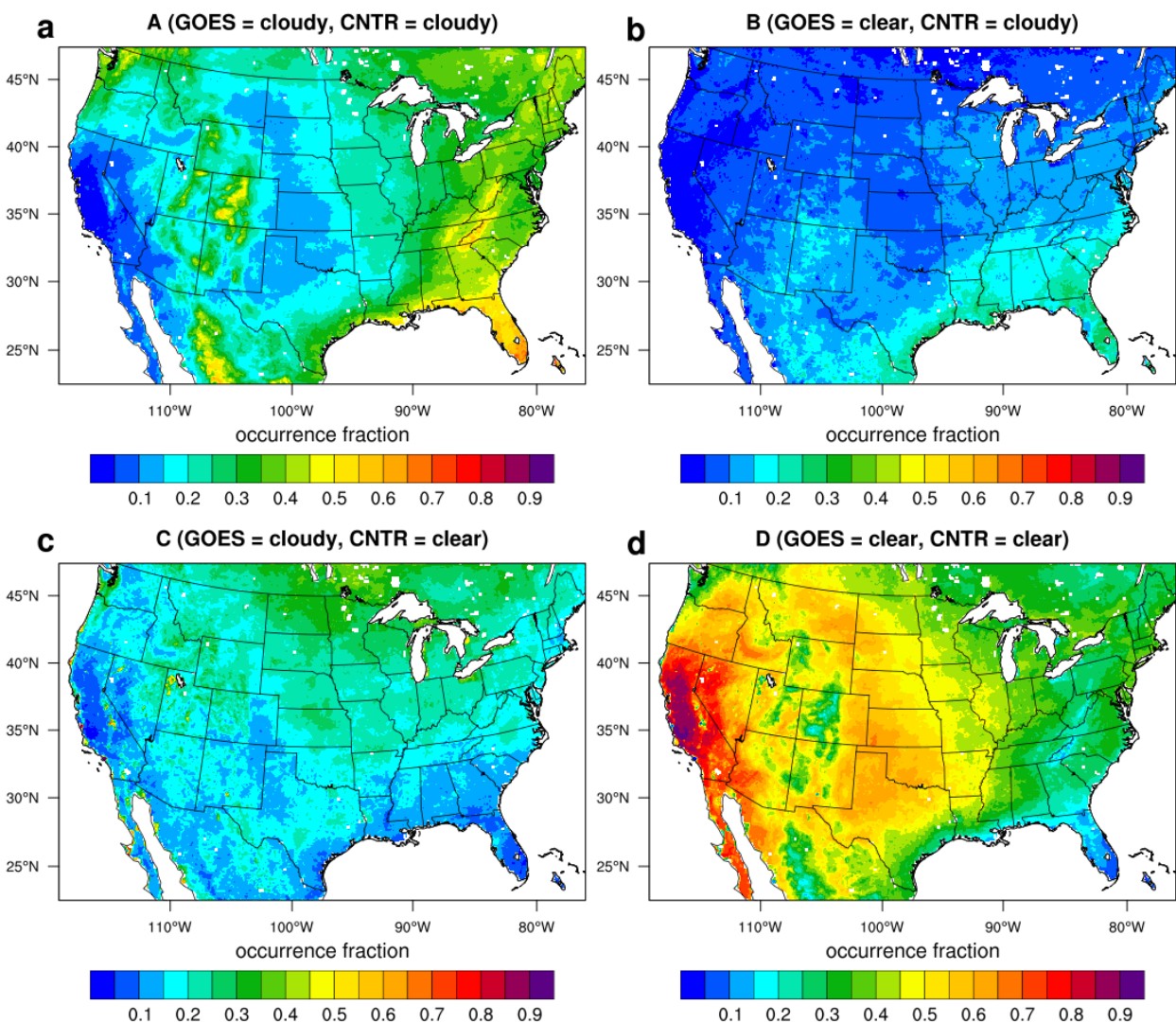


Fig. 1. Spatial distribution of each contingency category (see Table 2) between the WRF-generated clouds (CNTR simulation) and SatCORPS GOES retrievals averaged over the whole study period.




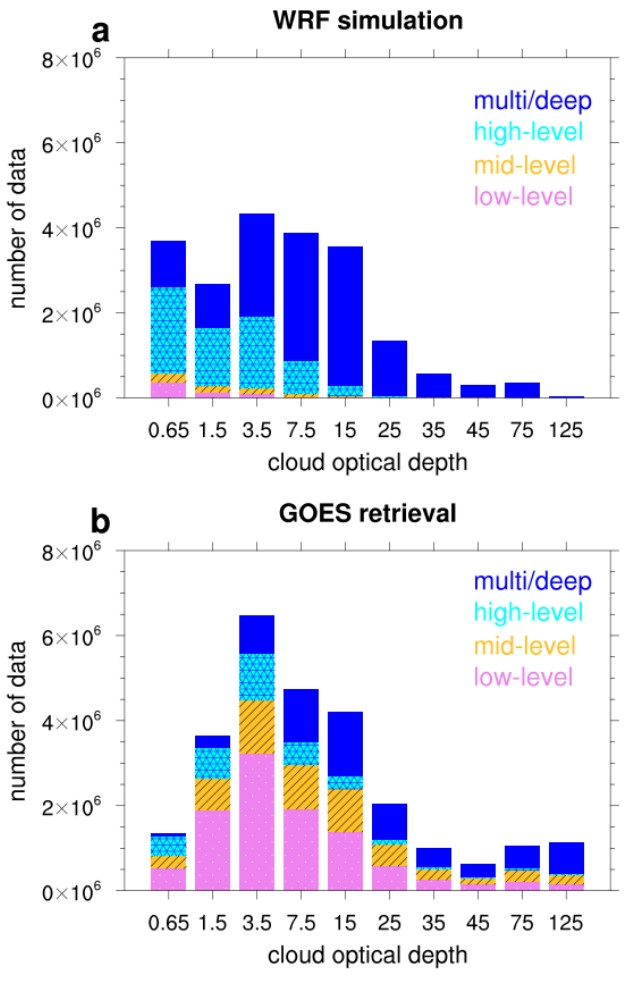


Fig. 2. Histogram of hourly cloud optical depth (COD) during the daytime (16–23 UTC) over CONUS (land only) from the (a) WRF simulation (with the Morrison microphysics and the Grell 3-D schemes) and (b) GOES satellite retrievals. CODs on the *x*-axis represent the mean values of the bins that are 0.3–1, 1–2, 2–5, 5–10, 10–20, 20–30, 30–40, 40–50, 50–100, and 100–150. For a fair comparison, the multi-layered WRF clouds are not resolved into cloud layers as this layering cannot be resolved by the satellite.









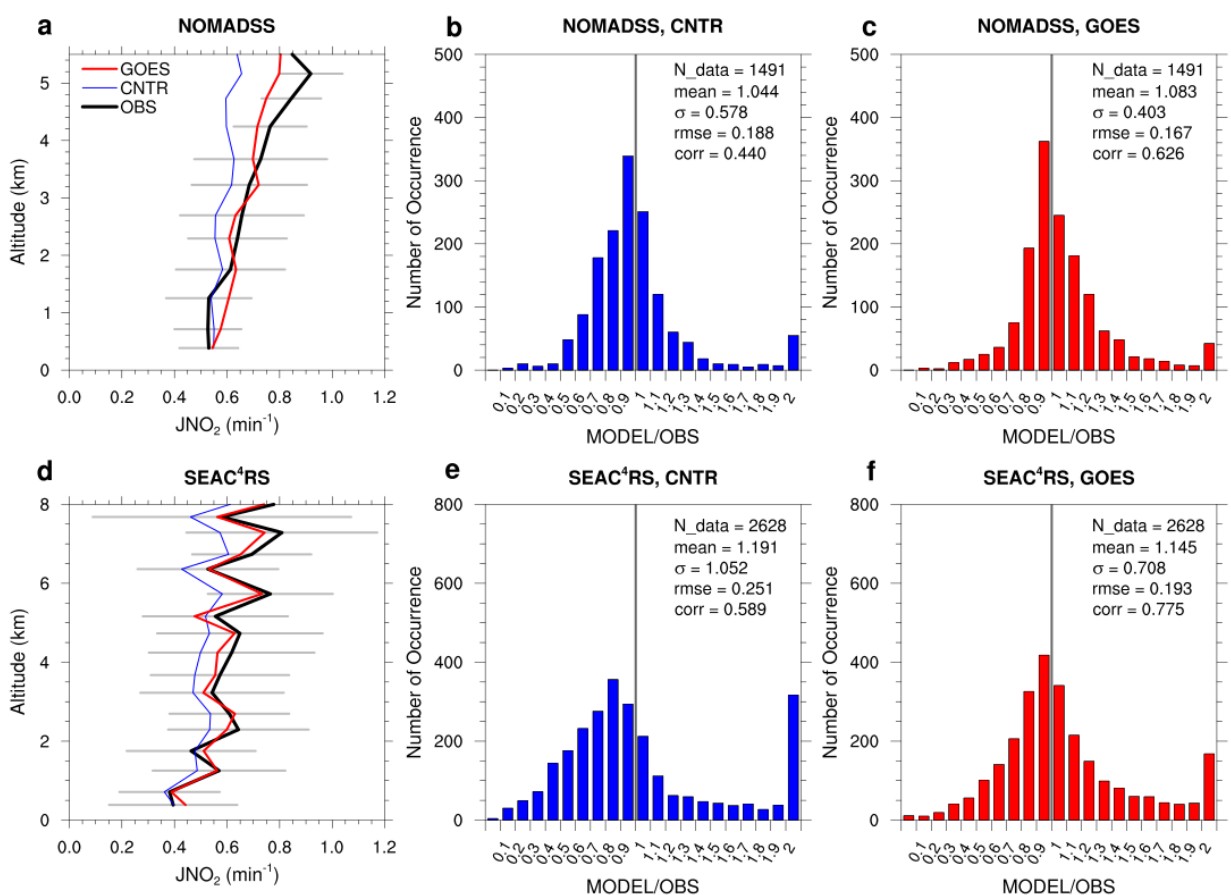


Fig. 3. Model evaluation with 16 NOMADSS flights (top row) and with 21 SEAC[4]RS flights

(bottom row). Note that only cloudy skies are considered. The comparison is performed for the

averaged vertical profiles of $JNO_2$ for the (a) NOMADSS and (d) SEAC[4]RS. The gray horizontal

lines indicate the standard deviations from the observations. Histogram of ratio of $JNO_2$

simulated by the model to $JNO_2$ observed (b) in the CNTR simulation and (c) in the GOES

simulation for the NOMADSS. (e and f) are the same as (b and c), respectively, but for the

SEAC[4]RS.

899

900

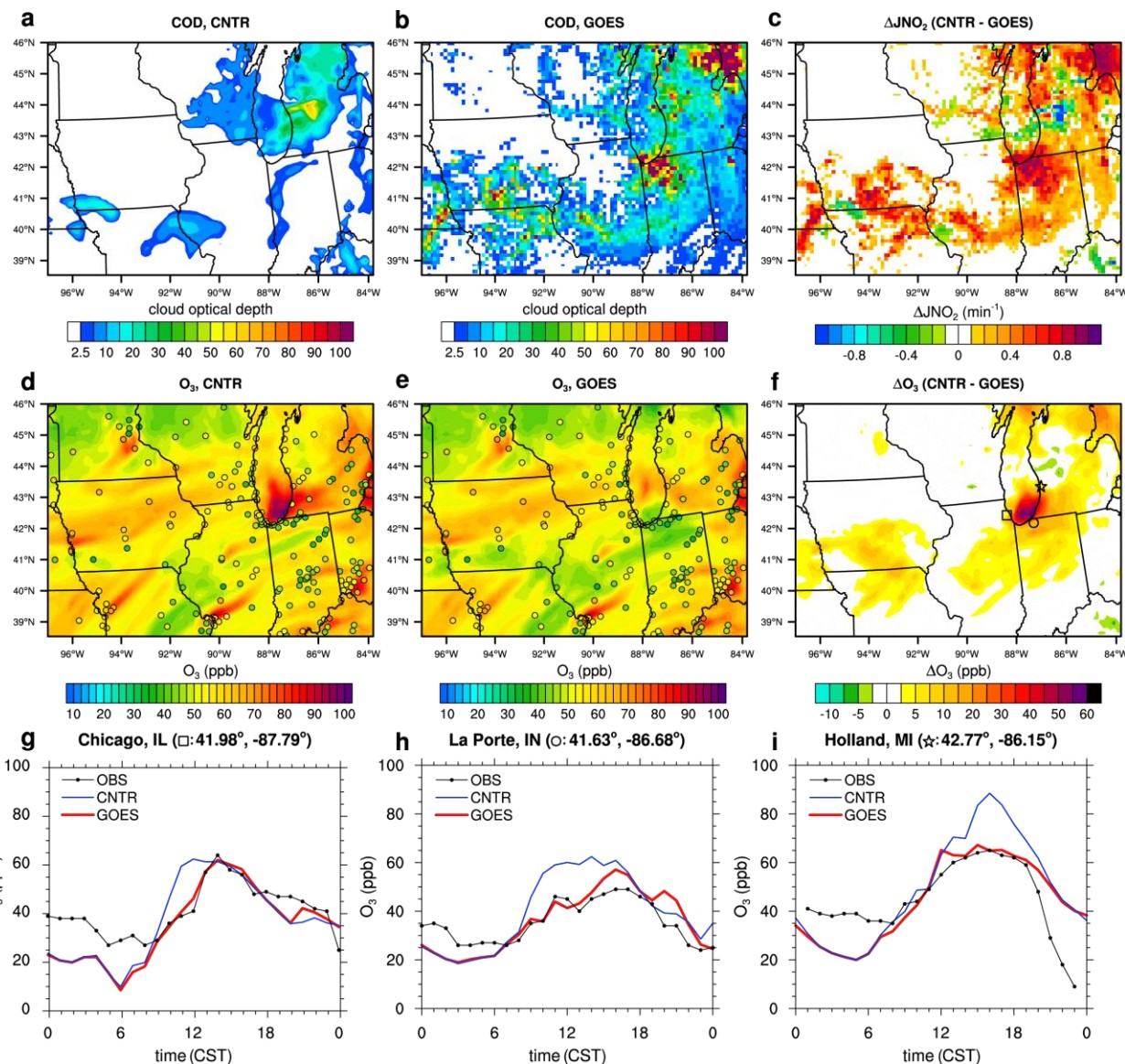

901

Fig. 4. Horizontal distributions of cloud optical depth at 13 CST (= 19 UTC) 8 July 2013 (a) in the CNTR simulation and (b) in the GOES simulation. Horizontal distributions of $O_3$ at 13 CST 8 July 2013 at the lowest model level (shaded) (d) in the CNTR simulation and (e) in the GOES simulation. The circles indicate EPA ozone measurements. (c and f) Difference in $JNO_2$ and $O_3$, respectively, between the simulations (i.e., CNTR simulation minus GOES simulation). (g, h, and i) Time series of $O_3$ at the square (Chicago, IL), circle (La Porte, IN), and star (Holland, MI) that are marked in (f), respectively.

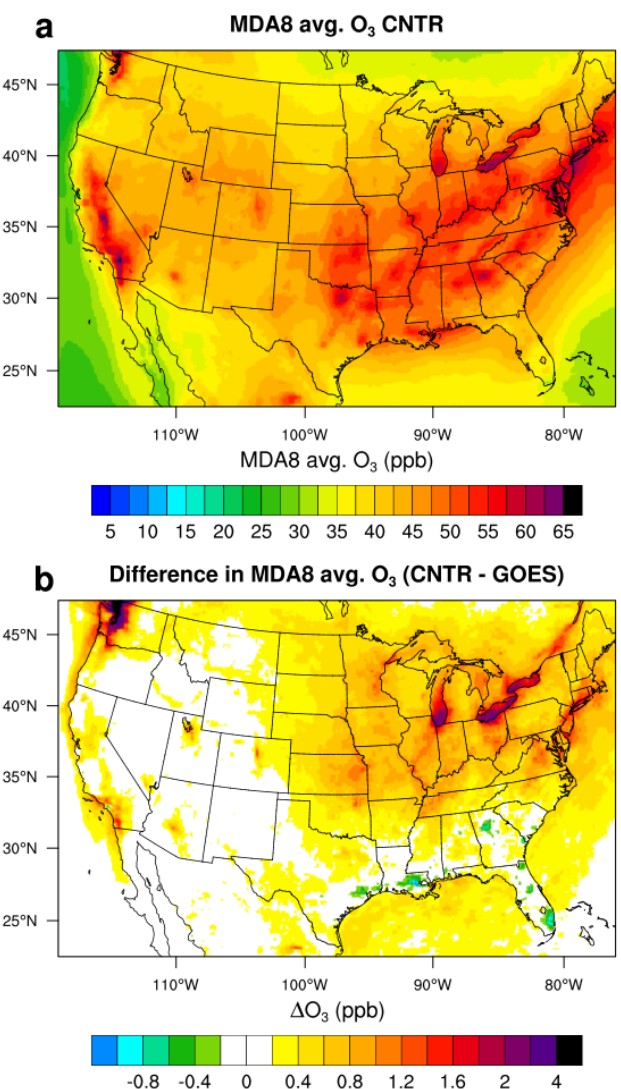

909

Fig. 5. (a) Spatial distribution of maximum daily 8-h average $O_3$ (MDA8 $O_3$) at the lowest model

level averaged over the whole analysis period in the CNTR simulation. (b) Difference in MDA8

$O_3$ at the lowest model level between the control and GOES simulations (i.e., CNTR minus

GOES).




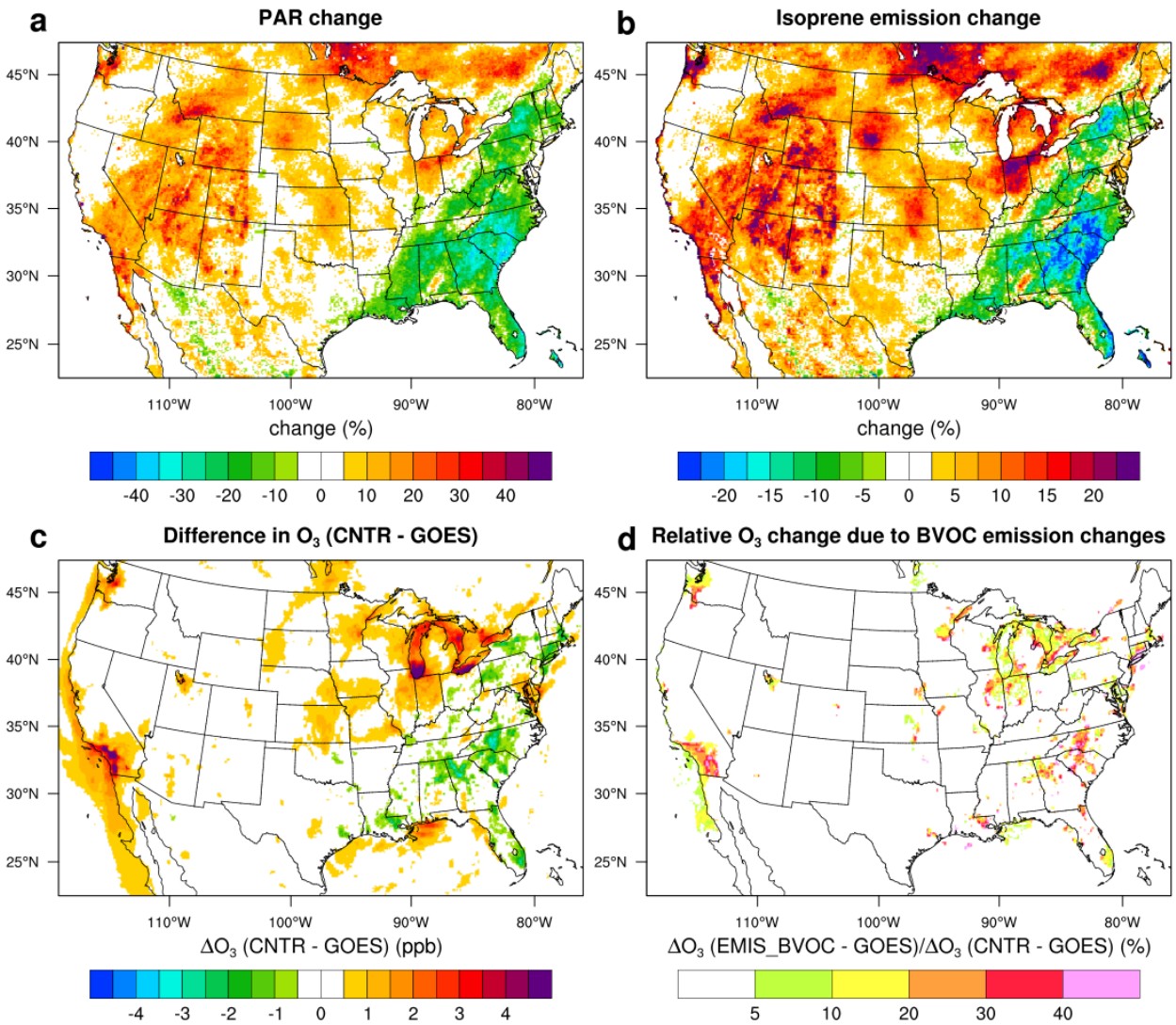


Fig. 6. Spatial distributions of (a) PAR change and (b) isoprene emission change from biogenic
sources between EMIS_BVOC and GOES simulations, (EMIS_BVOC–GOES)/GOES, averaged
over the period of 3–12 July 2013. (c) Difference in $O_3$ between the CNTR and GOES
simulations. (d) Ratio of $O_3$ difference between EMIS_BVOC and GOES simulations to $O_3$
difference between CNTR and GOES simulations, i.e., $\Delta O_3$(EMIS_BVOC–GOES)/$\Delta O_3$ (CNTR–
GOES). Note that the grids that have considerable $O_3$ difference between CNTR and GOES
simulations (> 1 ppb) are depicted in (d).

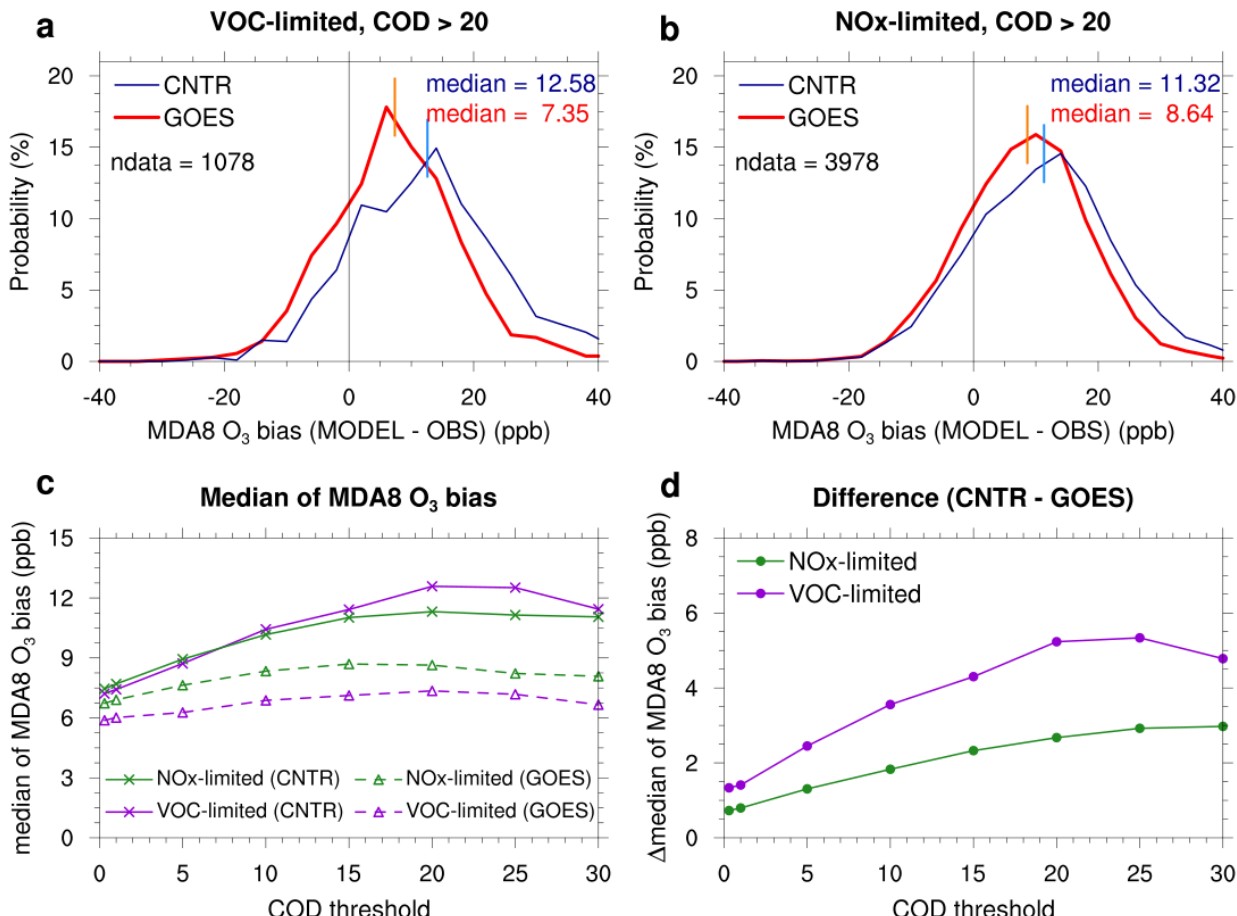

925

Fig. 7. (a) Probability density function of maximum daily 8-h average (MDA8) $O_3$ bias (model

value minus observation value) for VOC-limited regime under cloudy sky conditions defined

with COD threshold of 20. (b) Same as (a), but for $NO_X$-limited regime. (c) Median values of

MDA8 $O_3$ bias with respect to COD threshold in the CNTR simulation (solid lines with cross

marks) and in the GOES simulation (dashed line with triangles) for VOC-limited (purple color)

and $NO_X$-limited regimes (green color). (d) Difference in median values of MDA8 $O_3$ bias

between the two simulations with respect to COD threshold (i.e., CNTR minus GOES).

933

934

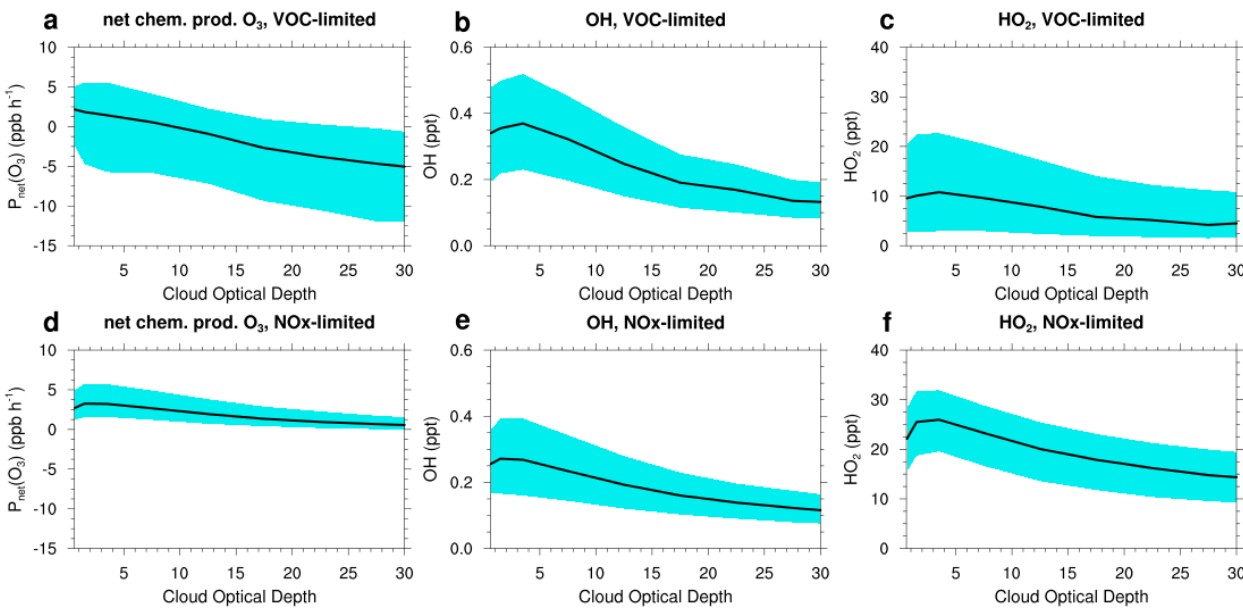

935

Fig. 8. (a) Net chemical production of $O_3$, (b) OH concentration, and (c) $HO_2$ concentration with variations of cloud optical depth for VOC-limited regime. The black line indicates the median and cyan shading indicates the 25 and 75 percentiles. Similar variables are shown for the $NO_X$-limited regimes (d, e, and f).






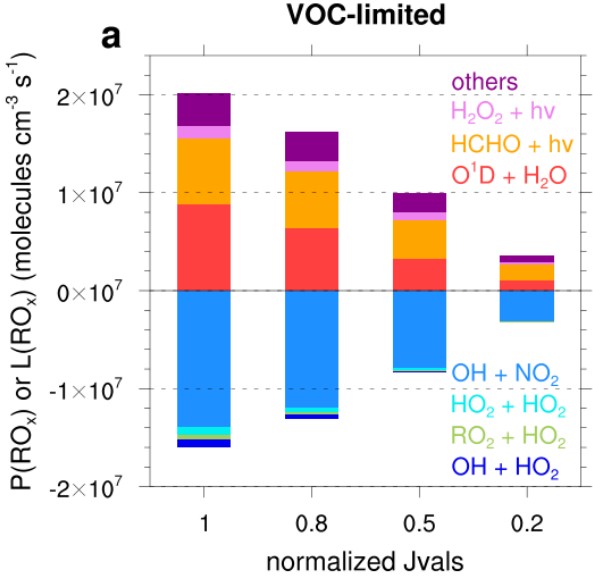

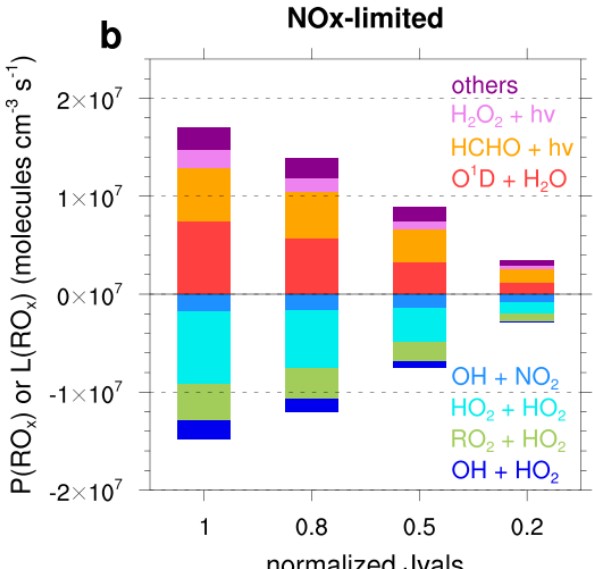


Fig. 9. Results of box modeling for production and loss rates of ROx (= OH + HO$_2$ + RO$_2$) radicals. "Others" in the legend indicates the photolysis of VOCs and reactions between alkenes and O$_3$. The value of 1 of normalized Jvals on $x$-axis indicates the photolysis rates for clear sky conditions.

950