# Peer review of "Quantifying errors in surface ozone predictions associated"

_Atmospheric Chemistry and Physics, 2017_

## Short Comment (SC1) · 6 Nov 2017

This is an interesting paper that suggests a possible explanation for typical model over-prediction of surface ozone over CONUS (Figure 7). It is not clear however if model simulations are improved both in terms of surface O3 predictions, as well as O3 vertical profiles (especially in the boundary layer and just above the boudary layer). While comparisons with measured vertical profiles of JNO2 are shown in Figure 3, no corresponding comparisons of vertical profiles are shown for O3. It would be useful to show these comparisons (and provide histograms as is done for JNO2) with simultaneous aircraft O3 measurements, especially given the overprediction of JNO2 in the boundary layer in the GOES simulation compared to the CNTR simulation for the NOMADSS flights (Figure 3).

In terms of model evaluation, it would be also useful to show comparisons of the modeled Ox vs NOz relationship against observations (as is done in Travis et al., 2016) as a check on modeled ozone production efficiency.

————————————————————

---

## Referee Comment (RC1) · Anonymous Referee #1 · 22 Nov 2017

This is a very good manuscript that demonstrates how the inclusion of satellite derived cloud properties can be used to improve atmospheric chemistry model simulations. I concur with Prasad Kasibhatla's comment that vertical profiles of O3, as well as NOx and NOz, should be evaluated with the field campaign observations. In addition, I wonder if the conclusions from the analyses involving NOx limited regimes are biased toward rural sites and therefore don't accurately represent NOx limited urban and sub-urban sites. A further analysis as discussed below could answer this question.

————————————

Major comments:

[Figure]

1) Evaluate vertical profiles of O3, NOx, NOz from field campaigns.

2) I have some reservations concerning the analyses involving NOx and VOC limited regimes in Sections 5.4 and 5.5 (although I like the last paragraph in Section 5.5). This manuscript has specific conclusions for VOC and NOx limited regimes. There are urban areas that are NOx limited. I suspect the NOx limited conclusions are heavily weighted toward rural areas and don't accurately represent polluted urban and suburban areas. I suggest binning sites based on ozone concentrations and then performing the analyses described in Sections 5.4 and 5.5 so the reader can compare VOC and NOx limited sites with similar ozone concentrations as well as VOC limited sites over a range of ozone concentrations and NOx limited sites over a range of ozone concentrations. Perhaps this can be done by binning the sites based on the peak maximum 8 hour average ozone concentration throughout the year (i.e., bin 1: peak MDA8>75, bin 2: peak MDAO3 between 70-75, . . .). It may be interesting to include the sites that fall into the transitional zone in your analysis. Include a figure showing delta O3 / delta NOy to identify NOx and VOC limited regimes.

————————————

Minor comments:

Abstract, line26: Remove mention of "robust with respect to the choice of the microphysics scheme." Only 2 microphysics schemes were tested.

Page 5, lines 89-91: Why skip pixels to create an 8km product? Why not leave the product at 4 km?

Page 9, line 181: Change "and with fire" to "and fire"

Page 10, line 189: Change "(Sillman and He (2002)" to "Sillman and He (2002)"

Page 11, lines 203-204: Change "wrong clouds (that are not present in reality)" to "clouds that are not present in reality"

Page 11, lines 204-205: Re-word this sentence.

Page 11, line 207: change "except for the mountain regions and northwestern US" to "except for parts of the Rocky Mountains and the Pacific Northwest."

Page 11, line 208: Change "in the central" to "in central"

Page 13, lines 252-253: Change "This is" to "These reductions are". Provide a further explanation of this claim.

Page 13, lines 254-260: This text states that NOMADSS has a larger mean model-to-observation ratio than SEAC4RS. This is not the case based on Figure 3.

Section 5.2: Calculate and discuss model-observations comparison statistics. Use maximum daily 8 hour average O3 (MDAO3) instead of 8hr average ozone between 10-17 LST.

Section 5.3: If you have a simulation with "photolysis with WRF clouds and PAR with GOES clouds", this would be interesting to include in this section.

Page 16, lines 316-318 and Figure 6: Difficult to see the relative differences between Figure 6c and 6d. A figure of the absolute value of 6d divided by the absolute value of 6c may be helpful.

Page 16, lines 318-320: Ozone difference of a simulation with photolysis with WRF clouds and PAR with GOES clouds minus GOES may or may not be 80% of CNTR-GOES. I suggest rewording this sentence to "The contribution of changes in BVOC emissions is $\sim$20% compared to changes of BVOC emissions and photolysis rates using GOES observations."

Figure 4: Use EST or CST, not LST. Map shows areas in the eastern and central time zone.

Figure 5: Show 3 panels with a CNTR, GOES, and difference plot (CNTR-GOES). Include observations overlayed on-top of the CNTR and GOES plots.

[Figure]

---

## Referee Comment (RC2) · Anonymous Referee #2 · 14 Dec 2017

This is a review of the manuscript titled "Quantifying errors in surface ozone predictions associated with clouds over CONUS: A WRF-Chem modeling study using satellite cloud retrievals". Overall the manuscript is well written and presents a very interesting analysis of integrating observed clouds into the WRF-Chem model to help correct errors in model simulated clouds. I agree with the comments already submitted, and only have a few additional comments to make regarding the manuscript. I don't see any reason to hold up publication of the manuscript once the comments/suggestions have been addressed.

General comments:

The authors mention several times in the manuscript the useful of this technique to improve ozone forecasts. I find this odd, since the technique described uses satellite observed clouds to correct model errors. How would this benefit forecasts? Is the assumption that these satellite data could be assimilated in near real-time, improving the near-term forecast of ozone? Some clarification seems necessary here to explain exactly what the authors have in mind for improving forecasts.

I'm also curious about the meteorological performance, although I realize that the cloud assimilation technique only applies to clouds as they affect photolysis. Since WRF tends to underpredict clouds in some regions and overpredict clouds in other regions, does that underprediction/overprediction manifest itself in the meteorological performance (e.g. surface temperature)? If so, this would imply to me that while assimilating clouds to improve photolysis is clearly important, improving clouds in WRF itself, and thereby hopefully improving the overall WRF performance, would be the ultimate goal, since surface temperature (and other meteorological variables), play an important role in not just ozone chemistry but in aerosol chemistry as well. More of thought than something that needs to be addressed in this article.

Specific comments:

Line 14: What is meant by "attributed to that in cloud predictions"?

Line 45: Is surface ozone hourly? Perhaps specify if it is.

Line 206: Change "over CONUS" to "over the CONUS".

Line 208: Remove "the" before central California.

Line 211: Change "in supplementary" to "in the supplementary material".

Line 288: I would be a little careful calling this 8-h average O3, since commonly 8-h average O3 refers to calculation of finding the maximum O3 across a number of 8-h averages throughout the day, whereas it appears the authors are simply using an afternoon average consisting of 8 hours. This might cause some confusion to some

readers.

Line 361: This should be changed to say "partially corrected". It would be presumptions to assume that the cloud fields have been fully corrected. It is a big step in the right direction though.

Line 410: Remove "relatively" before greater.

Fig 5. What is the cause of the very large reduction in O3 over the great lakes in the GOES simulation? Is that due to an improvement in clouds over the lakes themselves, or is it the result of improved clouds over the land and advection of O3 over the lakes? High O3 over the great lakes is a persistent problem in many air quality models, so the resulting improvement warrants some additional discussion in my opinion.
* * *

---

## Referee Comment (RC3) · Anonymous Referee #3 · 15 Dec 2017

This manuscript describes the results of WRF-Chem model simulations over the continental US at 12-km resolution in which the photolysis and biogenic emissions have been improved by substituting GOES satellite clouds for the clouds produced by the model itself. Significant improvement in the high bias for ozone prediction has been obtained. In general, the paper is well written, very readable, and the quality of the science is good.

However, there are two major issues that need to be addressed before it could be accepted: 1) The analysis is based primarily on one set of model physics (Morrison microphysics and Grell 3-D convection). The authors do test the sensitivity of the results

to a second microphysics scheme (Thompson) and found little difference. However, the simulation is for summer conditions (June to September), when a significant amount of cloudiness is due to convection. Therefore, there should be a sensitivity test also run with a second convective scheme. I would suggest running the relatively new Grell-Frietas scheme. From what I have seen, this scheme will produce more clouds.

2) In Section 2.3 the authors use the delta O3 to delta NOy ratio to determine VOC-limited and NOx-limited conditions. How is delta NOy determined at EPA monitoring sites? NOy is not routinely measured at these sites. Even true NOx is measured at only some small fraction of the O3 monitoring sites. This issue needs explanation or substantive revision.

Other more minor issues are as follows:

line 127: Which year NEI NOx was too high? Did Travis et al. indicate all NOx emission types were overestimated, or was it primarily mobile sources?

lines 255 to 260: I don't follow this description of cloud fraction. Please clarify.

Section 5.5 describes in detail how the box model calculations show that OH is less sensitive to changes in radiation in the NOx-limited regime. Some statements also need to be made about the effect on P(O3) in the box model.

---

## Author Comment (AC2) · 23 Mar 2018

**Responses to Reviewer 1's comments**

We thank the reviewer for providing valuable comments. We have improved our manuscript following his/her suggestions and comments. Please find our responses below. Reviewer's comments are highlight in blue.

**Major comments:**

C1) Evaluate vertical profiles of O3, NOX, NOZ from field campaigns.

Figure R1 shows the average vertical profiles of O3, NOX, HNO3 (top panels) and their rootmean-square-error (RMSE) from SEAC4RS campaign. NOZ species except for HNO3 are not saved in the WRF outputs along the flight tracks (at 1-min time intervals), and thus only HNO3 is compared here. The modeled vertical profiles of O3, NOX, and HNO3 are in a reasonable agreement with observations. The large deviations in  $O_3$  near the surface were also reported in previous studies such as Travis et al. (2016). The campaign average differences in vertical profiles of O3 between CNTR and GOES simulations are small as the aircraft measurements are mostly made in rural environments or high altitudes where  $O_3$  precursor concentrations are low. As shown in the manuscript, the effects of cloud correction are larger under high-NOX environments than low-NOX environments. However, it is seen that the cloud corrections slightly reduce  $O_3$  RMSE in general particularly below ~1 km altitude. Some examples from SEAC4RS and NOMADSS flights show that the effects of cloud correction can be considerable if the aircraft flew over relatively high-NOX regions under cloudy conditions (please see Figs. P3 and P4 in the responses to Dr. Kasibhatla's comments). Even though clouds were present during some flights, the cases allowing to estimate their effects are sparse as aircrafts usually avoid flying on heavily cloudy days. So, when all the data are averaged, the effects of cloud correction are expected to be small. The average profiles and RMSE of NOX and HNO3 from CNTR and GOES simulations are also very similar to each other.

Fig. R1. (Top, from left to right) Averaged vertical profiles of  $O_3$ ,  $NO_X$ , and  $HNO_3$ , respectively, for SEAC4RS measurements. The aircraft data over land within the southeast region (latitude: 25–40°N, longitude: 95–70°W) are only used for the averages. (Bottom, from left to right) The corresponding root-mean-square-error (RMSE) of  $O_3$ ,  $NO_X$ , and  $HNO_3$ , respectively.

C2) I have some reservations concerning the analyses involving NOx and VOC limited regimes in Sections 5.4 and 5.5 (although I like the last paragraph in Section 5.5). This manuscript has specific conclusions for VOC and NOx limited regimes. There are urban areas that are NOx limited. I suspect the NOx limited conclusions are heavily weighted toward rural areas and don't accurately represent polluted urban and suburban areas. I suggest binning sites based on ozone concentrations and then performing the analyses described in Sections 5.4 and 5.5 so the reader can compare VOC and NOx limited sites with similar ozone concentrations as well as VOC limited sites over a range of ozone concentrations and NOx limited sites over a range of ozone concentrations and NOx limited sites based on the peak maximum 8 hour average ozone concentration throughout the year (i.e., bin 1: peak MDA8>75, bin 2: peak MDAO3 between 70-75, ...). It may be interesting to include the sites that fall into the transitional zone in your analysis. Include a figure showing delta O3 / delta NOy to identify NOx and VOC limited regimes.

Thanks for providing these valuable suggestions. We agree that analyses for the VOC- and  $NO_{X}$ limited sites that have similar ranges of O3 concentration would provide more fair comparisons. Therefore, we performed additional analyses of the sensitivity of maximum daily 8-h average (MDA8) O3 bias to cloud correction in VOC- and NOX-limited regimes that have similar peak MDA8 O3 values. As O3 concentration is high in summertime, we only consider the period of June through September 2013. All the EPA sites are sorted into several bins based on peak (maximum) MDA8 O3 concentration during the period of June–September 2013. Figure R2 shows the same analysis as done in Fig. 7 but for various MDA8 bins. Please note that the COD threshold of 30 is not shown here because the number of data with this threshold is too small (generally less than ~50) when the sites are grouped into bins. It is clearly seen that the effects of cloud correction on reducing O3 bias are greater in VOC-limited regimes than NOX-limited regimes for all the bins although the degree is somewhat different among the bins. For the NOxlimited sites that have peak MDA8  $O_3 > 75$  ppb, the maximum decrease in  $O_3$  bias due to cloud correction is ~3.5 ppb and this value is similar to that (~3 ppb) found in the analysis for all the sites (Fig. 7d in the manuscript). The NOX-limited sites with peak MDA8  $O_3 > 75$  ppb are mostly located near the major US cities or the state of California (Fig. R3). Those sites are likely characterized by polluted urban or suburban areas. For the NOX-limited sites with peak MDA8 O3 of 60–65 ppb that are mostly located in rural environments, for example, the effects of cloud correction on reducing  $O_3$  bias (maximum value of ~2 ppb) are smaller than those seen for the sites with peak MDA8  $O_3 > 70$  ppb (maximum value of ~4 ppb). So, even for NOx-limited regimes it can be said that the effects of cloud correction are larger in more polluted areas. Still, however, the effects of cloud correction are larger in VOC-limited regimes than NOX-limited regimes. Therefore, our conclusions originally drawn in the manuscript remain unchanged. We mentioned the results of this analysis in the manuscript as follows.

"We performed additional analysis by dividing VOC- and NOX-limited sites into groups that have similar ranges of peak MDA8  $O_3$  concentration during the period of June–September 2013 (Fig. S3). All sites are grouped into bins with peak value of MDA8  $O_3$  ranging from larger than 75 ppb, 70–75 ppb, 65–70 ppb, 60–65 ppb, to smaller than 60 ppb. The maximum reduction in  $O_3$  bias due to cloud corrections is obtained for the VOC-limited sites with peak MDA8  $O_3$  of 65–70 ppb and reaches ~8 ppb. The maximum reduction for NOX-limited sites, on the other hand, is ~4 ppb and is found for the sites with peak MDA8  $O_3$  of 70–75 ppb. Although the degree of the  $O_3$  bias reduction varies somewhat among the bins for a given ozone regime, the effects of cloud correction on  $O_3$  bias reduction remain larger in VOC-limited regimes than NOX-limited regimes."

---

## Author Comment (AC3) · 23 Mar 2018

**Responses to Reviewer 3's comments**

Thank you so much for providing valuable comments. We have improved our manuscript following your suggestions and comments. Please find our responses below. Your comments are highlight in blue.

C1) The analysis is based primarily on one set of model physics (Morrison microphysics and Grell 3-D convection). The authors do test the sensitivity of the results to a second microphysics scheme (Thompson) and found little difference. However, the simulation is for summer conditions (June to September), when a significant amount of cloudiness is due to convection. Therefore, there should be a sensitivity test also run with a second convective scheme. I would suggest running the relatively new Grell-Freitas scheme. From what I have seen, this scheme will produce more clouds.

Following reviewer's suggestion, we have performed sensitivity tests with Grell-Freitas scheme. As done for microphysics scheme, a period of 10 days (3–12 July 2013) was considered. An example showing spatial distribution of cloud optical depth from the two cumulus parameterization schemes are presented in Fig. T1. In general, the spatial patterns and the location of large systems are similar to each other. The Grell-Freitas scheme produces more and/or thicker clouds in some regions such as the north Michigan and the south Ohio than the Grell-3D scheme. However, the Grell-Freitas scheme produces fewer and/or thinner clouds in other regions such as the east Texas and North Carolina. In Fig. T2, the histograms of cloud optical depth obtained for the 10-day period from Grell-Freitas scheme (left) and from Grell-3D scheme (right) show that the distributions of cloud optical depth are in general similar to each other. The Grell-Freitas scheme tends to produce fewer clouds with small or moderate cloud optical depth. Figure T3 shows that the degree of cloud correction in reducing $O_3$ bias is larger in VOC-limited regimes than in $NO_X$-limited regimes in the simulation with Grell-Freitas scheme, and thus the conclusions originally drawn remain unchanged.

We included the summary of this discussion above in the revised manuscript and figures (Figs. T2 and T3) in the supplementary materials.

[Figure]

Fig. T1. Cloud optical depth (COD) at 19 UTC 8 July 2013 using the (left) Grell-Freitas scheme and (right) Grell-3D scheme. Both simulations use the Morrison microphysics scheme.

[Figure]

Fig. T2. Histogram of hourly cloud optical depths during the daytime (16–23 UTC) over CONUS (land only) for the period of 3–12 July 2013 from simulations with the (left) Grell-Freitas scheme and (right) Grell-3D scheme.

[Figure]

Fig. T3. (Left column) The results of 3–12 July 2013 WRF-Chem simulations with Grell-Freitas scheme. (a/c) Probability density function of MDA8 $O_3$ bias (model value minus observation value) for VOC/$NO_X$-limited regime under cloudy sky conditions defined with COD threshold of 20 in the simulations with the Grell-Freitas scheme. (b/d) Same as (a/c), but for the simulations with the Grell-3D scheme. (e and f) Difference in median values of MDA8 $O_3$ bias between the two simulations with respect to COD threshold (i.e., CNTR minus GOES) for the simulations with the Grell-Freitas and with the Grell-3D schemes, respectively.

C2) In Section 2.3 the authors use the delta O3 to delta NOy ratio to determine VOC-limited and NOx-limited conditions. How is delta NOy determined at EPA monitoring sites? NOy is not

routinely measured at these sites. Even true NOx is measured at only some small fraction of the O3 monitoring sites. This issue needs explanation or substantive revision.

$NO_y$ used in this study is the modeled $NO_y$ and $O_3$ is also modeled $O_3$. As you indicated, $NO_y$ is not routinely measured, so the sites having $NO_y$ measurements are very limited. Therefore, we could not rely on $NO_y$ observations. We included the following sentence in the revised manuscript.

"*Note that modeled $O_3$ and $NO_y$ in the CNTR simulation are used to determine whether an EPA site is in VOC-limited or $NO_X$-limited regime because $NO_y$ measurements are available for limited sites*."

In addition, we included examples showing how to determine VOC-limited or $NO_X$-limited sites in the supplementary materials (Fig. S1).

Minor comments:
C3) line 127: Which year NEI NOx was too high? Did Travis et al. indicate all NOx emission types were overestimated, or was it primarily mobile sources?

Travis et al. (2016) used 2011 NEI emissions and adjusted to 2013. They reduced NOx emissions from mobile and industrial sources (all sources except for power plants). Based on the references mentioned in Travis et al. (2016), several local studies reported that NEI NOx emissions for mobile sources are high by a factor of 2 or more (Castellanos et al, 2011; Fujita et al., 2012; Brioude et al., 2013; Anderson et al., 2014).

In our present study, we reduced $NO_X$ emission from all anthropogenic sources by 40% based on the analysis of Travis et al. (2016), and this is mentioned in the revised manuscript.

Reference
Castellanos, P. Marufu, L. T., Doddridge, B. G., Taubman, B. F., Schwab, J. J., Hains, J. C., Ehrman, S. H., and Dickerson, R. R.: Ozone, oxides of nitrogen, and carbon monoxide during pollution events over the eastern United States: An evaluation of emissions and vertical mixing, J. Geophys. Res., 116, D16307, doi:10.1029/2010JD014540, 2011.
Fujita, E. M., Campbell, D. E., Zielinska, B., Chow, J. C., Lind-hjem, C. E., DenBleyker, A., Bishop, G. A., Schuchmann, B. G., Stedman, D. H., and Lawson, D. R.: Comparison of the MOVES2010a, MOBILE6.2, and EMFAC2007 mobile source emission models with on-road traffic tunnel and remote sensing measurements, J. Air Waste Manage., 62, 1134–1149, doi:10.1080/10962247.2012.699016, 2012.
Brioude, J., Angevine, W. M., Ahmadov, R., Kim, S.-W., Evan, S., McKeen, S. A., Hsie, E.-Y., Frost, G. J., Neuman, J. A., Pollack, I. B., Peischl, J., Ryerson, T. B., Holloway, J., Brown, S. S., Nowak, J. B., Roberts, J. M., Wofsy, S. C., Santoni, G. W., Oda, T., and Trainer, M.: Top-down estimate of surface flux in the Los Angeles Basin using a mesoscale inverse modeling technique: as-sessing anthropogenic emissions of CO, NOx and CO2 and their impacts, Atmos. Chem. Phys., 13, 3661–3677, doi:10.5194/acp-13-3661-2013, 2013.

Anderson, D. C., Loughner, C. P., Diskin, G., Weinheimer, A., Canty, T., P., Salawitch, R. J., Worden, H. M., Fried, A., Mikoviny, T., Wisthaler, A., and Dickerson, R. R.: Measured and modeled CO and NOy in DISCOVER-AQ: An evaluation of emissions and chemistry over the eastern US, Atmos. Environ., 96, 78–87, doi:10.1016/j.atmosenv.2014.07.004, 2014.

C4) lines 255 to 260: I don't follow this description of cloud fraction. Please clarify.
This part originally explained the results without showing figures that are relevant to the cloud fraction, but without showing figures we concluded that this part was too confusing to reader, and we decided to remove it. Please see the comment 12 of the first reviewer and our responses.

C5) Section 5.5 describes in detail how the box model calculations show that OH is less sensitive to changes in radiation in the NOx-limited regime. Some statements also need to be made about the effect on P(O3) in the box model.
Figure T4 shows the net chemical production of $O_3$ in the box model, and the result is consistent with that is found in the WRF-Chem simulations: larger sensitivity of $P(O_3)$ to cloudiness in VOC-limited regimes than $NO_X$-limited regimes. We briefly included this result in the revised manuscript as follows.
*"Note that the net chemical production of $O_3$ obtained from the box model results also shows a larger sensitivity to cloudiness in VOC-limited regimes than in $NO_X$-limited regimes (not shown)."*

[Figure]

Fig. T4. The net chemical production of $O_3$ from the box model simulations.

---

## Author Response (AR1)

**Responses to Reviewer 1's comments**

We thank the reviewer for providing valuable comments. We have improved our manuscript following his/her suggestions and comments. Please find our responses below. Reviewer's comments are highlight in blue.

Major comments:

C1) Evaluate vertical profiles of $O_3$, $NO_X$, $NO_Z$ from field campaigns.

Figure R1 shows the average vertical profiles of $O_3$, $NO_X$, $HNO_3$ (top panels) and their root-mean-square-error (RMSE) from SEAC$^4$RS campaign. $NO_Z$ species except for $HNO_3$ are not saved in the WRF outputs along the flight tracks (at 1-min time intervals), and thus only $HNO_3$ is compared here. The modeled vertical profiles of $O_3$, $NO_X$, and $HNO_3$ are in a reasonable agreement with observations. The large deviations in $O_3$ near the surface were also reported in previous studies such as Travis et al. (2016). The campaign average differences in vertical profiles of $O_3$ between CNTR and GOES simulations are small as the aircraft measurements are mostly made in rural environments or high altitudes where $O_3$ precursor concentrations are low. As shown in the manuscript, the effects of cloud correction are larger under high-$NO_X$ environments than low-$NO_X$ environments. However, it is seen that the cloud corrections slightly reduce $O_3$ RMSE in general particularly below ~1 km altitude. Some examples from SEAC$^4$RS and NOMADSS flights show that the effects of cloud correction can be considerable if the aircraft flew over relatively high-$NO_X$ regions under cloudy conditions (please see Figs. P3 and P4 in the responses to Dr. Kasibhatla's comments). Even though clouds were present during some flights, the cases allowing to estimate their effects are sparse as aircrafts usually avoid flying on heavily cloudy days. So, when all the data are averaged, the effects of cloud correction are expected to be small. The average profiles and RMSE of $NO_X$ and $HNO_3$ from CNTR and GOES simulations are also very similar to each other.

[Figure]

Fig. R1. (Top, from left to right) Averaged vertical profiles of $O_3$, $NO_X$, and $HNO_3$, respectively, for SEAC[4]RS measurements. The aircraft data over land within the southeast region (latitude: 25–40°N, longitude: 95–70°W) are only used for the averages. (Bottom, from left to right) The corresponding root-mean-square-error (RMSE) of $O_3$, $NO_X$, and $HNO_3$, respectively.

C2) I have some reservations concerning the analyses involving NOx and VOC limited regimes in Sections 5.4 and 5.5 (although I like the last paragraph in Section 5.5). This manuscript has specific conclusions for VOC and NOx limited regimes. There are urban areas that are NOx limited. I suspect the NOx limited conclusions are heavily weighted toward rural areas and don't accurately represent polluted urban and suburban areas. I suggest binning sites based on ozone concentrations and then performing the analyses described in Sections 5.4 and 5.5 so the reader can compare VOC and NOx limited sites with similar ozone concentrations as well as VOC limited sites over a range of ozone concentrations and NOx limited sites over a range of ozone concentrations. Perhaps this can be done by binning the sites based on the peak maximum 8 hour average ozone concentration throughout the year (i.e., bin 1: peak MDA8>75, bin 2: peak MDAO3 between 70-75, …). It may be interesting to include the sites that fall into the transitional zone in your analysis. Include a figure showing delta O3 / delta NOy to identify NOx and VOC limited regimes.

Thanks for providing these valuable suggestions. We agree that analyses for the VOC- and $NO_X$-limited sites that have similar ranges of $O_3$ concentration would provide more fair comparisons. Therefore, we performed additional analyses of the sensitivity of maximum daily 8-h average (MDA8) $O_3$ bias to cloud correction in VOC- and $NO_X$-limited regimes that have similar peak MDA8 $O_3$ values. As $O_3$ concentration is high in summertime, we only consider the period of June through September 2013. All the EPA sites are sorted into several bins based on peak (maximum) MDA8 $O_3$ concentration during the period of June–September 2013. Figure R2 shows the same analysis as done in Fig. 7 but for various MDA8 bins. Please note that the COD threshold of 30 is not shown here because the number of data with this threshold is too small (generally less than ~50) when the sites are grouped into bins. It is clearly seen that the effects of cloud correction on reducing $O_3$ bias are greater in VOC-limited regimes than $NO_X$-limited regimes for all the bins although the degree is somewhat different among the bins. For the $NO_X$-limited sites that have peak MDA8 $O_3 > 75$ ppb, the maximum decrease in $O_3$ bias due to cloud correction is ~3.5 ppb and this value is similar to that (~3 ppb) found in the analysis for all the sites (Fig. 7d in the manuscript). The $NO_X$-limited sites with peak MDA8 $O_3 > 75$ ppb are mostly located near the major US cities or the state of California (Fig. R3). Those sites are likely characterized by polluted urban or suburban areas. For the $NO_X$-limited sites with peak MDA8 $O_3$ of 60–65 ppb that are mostly located in rural environments, for example, the effects of cloud correction on reducing $O_3$ bias (maximum value of ~2 ppb) are smaller than those seen for the sites with peak MDA8 $O_3 > 70$ ppb (maximum value of ~4 ppb). So, even for $NO_X$-limited regimes it can be said that the effects of cloud correction are larger in more polluted areas. Still, however, the effects of cloud correction are larger in VOC-limited regimes than $NO_X$-limited regimes. Therefore, our conclusions originally drawn in the manuscript remain unchanged. We mentioned the results of this analysis in the manuscript as follows.

*"We performed additional analysis by dividing VOC- and $NO_X$-limited sites into groups that have similar ranges of peak MDA8 $O_3$ concentration during the period of June–September 2013 (Fig. S3). All sites are grouped into bins with peak value of MDA8 $O_3$ ranging from larger than 75 ppb, 70–75 ppb, 65–70 ppb, 60–65 ppb, to smaller than 60 ppb. The maximum reduction in $O_3$ bias due to cloud corrections is obtained for the VOC-limited sites with peak MDA8 $O_3$ of 65–70 ppb and reaches ~8 ppb. The maximum reduction for $NO_X$-limited sites, on the other hand, is ~4 ppb and is found for the sites with peak MDA8 $O_3$ of 70–75 ppb. Although the degree of the $O_3$ bias reduction varies somewhat among the bins for a given ozone regime, the effects of cloud correction on $O_3$ bias reduction remain larger in VOC-limited regimes than $NO_X$-limited regimes."*

[Figure]

Fig. R2. Similar to Fig. 7 in the revised manuscript but for several bins with different peak MDA8 O$_3$ ranges.

[Figure]

Fig. R3. Maps showing the sites that belong to each peak MDA8 O$_3$ bin. The number in parenthesis indicate the number of sites in each ozone range. For example, the number of VOC-limited sites with peak MDA8 O$_3$ > 75 ppb is 119.

In addition, the sites that fall into the transitional zone are added in the analysis (Fig. R4). The effects of cloud correction on O$_3$ bias reduction for the transition sites are in-between those for the VOC-limited regimes and NO$_X$-limited regimes. This is now explained in the revised manuscript.

"*Note that the results for the sites in transitional zone (the slope of $\Delta O_3/\Delta NO_y$ is 4–6) showed that the effects of cloud in the transitional zone are intermediate; that is, larger than those for $NO_X$-limited regimes but smaller than those for VOC-limited regimes (not shown).*"

[Figure]

Fig. R4. Same as in Fig. 7 but with the results for transitional zone.

Examples of scatter plots of $O_3$ and $NO_y$, which are used to identify VOC- or $NO_X$-limited sites, are shown in Fig. R5 and following the reviewer's comment we included this in the supplementary material (Fig. S1).

[Figure]

Fig. R5. Scatter plots of $O_3$ and $NO_y$. The thick black line indicates the linear regression coefficient. The modeled $O_3$ and $NO_y$ concentrations at 15–16 local time under clear sky conditions (hourly COD < 1) in the CNTR simulation are used for analysis. On the title heading, the first and second words indicate the state and the county of the site. The third one indicates the type of the site defined by EPA.

Minor comments:
C3) Abstract, line26: Remove mention of "robust with respect to the choice of the microphysics scheme." Only 2 microphysics schemes were tested.
We have removed that part following the reviewer's comment.

C4) Page 5, lines 89-91: Why skip pixels to create an 8km product? Why not leave the product at 4 km?

The 8-km products are sampled every other pixel (4-km pixel) to save processing time. It does not affect the statistics of the analysis.

C5) Page 9, line 181: Change "and with fire" to "and fire"
It is changed.

C6) Page 10, line 189: Change "(Sillman and He (2002)" to "Sillman and He (2002)"
It is corrected.

C7) Page 11, lines 203-204: Change "wrong clouds (that are not present in reality)" to "clouds that are not present in reality"
It is changed.

C8) Page 11, lines 204-205: Re-word this sentence.
It is revised as follows.
"*The overall bias, (A+B)/(A+C), is 0.789 and this means that the WRF underestimates the frequency of cloudy skies.*"

C9) Page 11, line 207: change "except for the mountain regions and northwestern US" to "except for parts of the Rocky Mountains and the Pacific Northwest."
It is changed.

C10) Page 11, line 208: Change "in the central" to "in central"
It is changed.

C11) Page 13, lines 252-253: Change "This is" to "These reductions are". Provide a further explanation of this claim.
It is changed following the reviewer's suggestion. This claim was based on the histograms separating the cloud conditions into below, above, and inside cloud conditions (Fig. R6), which are not shown in the manuscript. The reductions of larger errors with model-to-observation ratio of greater than 2 are due to the reductions under below- and inside-cloud conditions. We elaborate the reasons in the revised manuscript as follows.
"*This is because the number of data influenced by considerably thick clouds is larger in SEAC$^4$RS than in NOMADSS and the measurements in the presence of those thick clouds were mostly made under below-cloud or inside-cloud conditions.*"

[Figure]

Fig. R6. Histogram of model-to-observation $JNO_2$ ratio for SEAC4RS under (top) below, (middle) above, and (bottom) inside cloud conditions.

C12) Page 13, lines 254-260: This text states that NOMADSS has a larger mean model-to-observation ratio than SEAC4RS. This is not the case based on Figure 3.

The text was intended to indicate the above cloud conditions. As Fig. R7 shows, the performance in the GOES simulation is not greatly improved even though the satellite clouds are used. The effects of cloud correction for above-cloud conditions for NOMADSS are different from those for SEAC[4]RS (Fig. R6 middle row). Given that the histograms of Fig. R7 are not included in the manuscript, we have deleted this part in the revised manuscript to avoid confusion.

[Figure]

Fig. R7. Histogram of model-to-observation JNO₂ ratio for NOMADSS under (top) below, (middle) above, and (bottom) inside cloud conditions.

C13) Section 5.2: Calculate and discuss model-observations comparison statistics. Use maximum daily 8 hour average O₃ (MDAO3) instead of 8hr average ozone between 10-17 LST.

We have added a discussion of the statistics in the manuscript as requested by the reviewer. Indeed, the root-mean-square-error (RMSE) and correlation coefficient are compared. Both the RMSE and correlation coefficient show a better performance when satellite clouds are used (GOES simulation) than when model clouds are used (CNTR simulation). The RMSE of MDA8 O₃ in the GOES (CNTR) simulation is 13.2 ppb (16.9 ppb) and the correlation coefficient of MDA8 O₃ in the GOES (CNTR) simulation is 0.5 (0.4). This is now explained in the manuscript:
*"The performance of the GOES simulation is found to be better than that of the CNTR simulation as compared to observations: for example, under cloudy conditions (COD > 20, see section 5.4 for the criterion), the root-mean-square error of MDA8 O₃ in the GOES (CNTR) simulation is 13.2 ppb (16.9 ppb) and the correlation coefficient of MDA8 O₃ in the GOES (CNTR) simulation is 0.5 (0.4)."*

The spatial ozone distribution map averaged over the study period (Fig. 5) is replaced with MDA8 $O_3$ in the revised manuscript. The result with MDA8 $O_3$ is very similar to that shown with daytime 8-h average (10–17 LST) $O_3$.

C14) Section 5.3: If you have a simulation with "photolysis with WRF clouds and PAR with GOES clouds", this would be interesting to include in this section.
We agree with that. Unfortunately, the current model does not have capability to simulate the setup proposed by the reviewer.

C15) Page 16, lines 316-318 and Figure 6: Difficult to see the relative differences between Figure 6c and 6d. A figure of the absolute value of 6d divided by the absolute value of 6c may be helpful.
Following the reviewer's suggestion, we replaced Fig. 6d with a plot showing the ratio of difference in $O_3$ between EMIS_BVOC and GOES (previously Fig. 6d) to difference in $O_3$ between CNTR and GOES (Fig. 6c). The description of this figure is as follows.
"*Figure 6d shows the relative $O_3$ difference between EMIS_BVOC and GOES simulations to $O_3$ difference between CNTR and GOES simulations (Fig. 6c).*"

C16) Page 16, lines 318-320: Ozone difference of a simulation with photolysis with WRF clouds and PAR with GOES clouds minus GOES may or may not be 80% of CNTR-GOES. I suggest rewording this sentence to "The contribution of changes in BVOC emissions is ~20% compared to changes of BVOC emissions and photolysis rates using GOES observations."
It is revised based on the reviewer's suggestion:
"*The average contribution of changes in BVOC emissions over land is ~20% compared to changes of BVOC emissions plus photolysis rates using GOES satellite clouds.*"

C17) Figure 4: Use EST or CST, not LST. Map shows areas in the eastern and central time zone.
LST is changed to CST.

C18) Figure 5: Show 3 panels with a CNTR, GOES, and difference plot (CNTR-GOES). Include observations overlayed on-top of the CNTR and GOES plots.
The reason why we show only the result of CNTR simulation is that the spatial distribution of average $O_3$ in GOES simulation is similar to that in CNTR simulation although the ozone levels are different (Fig. R8). We mentioned the reason why the result of GOES simulation is not shown here in the revised manuscript. In addition, adding observations on the map makes the plot very complicated as the number of sites is ~1300. Lots of sites are closely located to each other as shown in Fig. R3. When all the sites in the bins in Fig. R3 are plotted and overlayed on a map, readers will not be able to see the values of observations.

[Figure]

Fig. R8. Spatial distribution of MDA8 $O_3$ at the lowest model level averaged over the study period (top, left) in the CNTR simulation and (top, right) in the GOES simulation. (Bottom, left) Difference in MDA8 $O_3$ between CNTR and GOES simulations.

**Responses to Reviewer 2's comments**

Authors thank the reviewer for providing valuable comments. We have improved our manuscript to address his/her suggestions. Reviewer's comments are highlight in blue, and our responses are in black.

General comments:
C1) The authors mention several times in the manuscript the useful of this technique to improve ozone forecasts. I find this odd, since the technique described uses satellite observed clouds to correct model errors. How would this benefit forecasts? Is the assumption that these satellite data could be assimilated in near real-time, improving the near-term forecast of ozone? Some clarification seems necessary here to explain exactly what the authors have in mind for improving forecasts.

The goal of our study is to quantify the potential benefit of improved cloud fields in air quality modeling. We did not estimate by how much the near real-time ozone forecast could be improved using the observed clouds through data assimilation. This will be done in a future study. The conclusion has been modified to avoid any confusion:

*"From the perspective of $O_3$ forecast, our study indicates that there is a need for an enhanced understanding of the evolution of errors in $O_3$ forecasts associated with errors in cloud forecasts, and for optimizing the use of meteorological forecasts to allow more accurate near-term $O_3$ predictions."*

C2) I'm also curious about the meteorological performance, although I realize that the cloud assimilation technique only applies to clouds as they affect photolysis. Since WRF tends to underpredict clouds in some regions and overpredict clouds in other regions, does that underprediction/overprediction manifest itself in the meteorological performance (e.g. surface temperature)? If so, this would imply to me that while assimilating clouds to improve photolysis is clearly important, improving clouds in WRF itself, and thereby hopefully improving the overall WRF performance, would be the ultimate goal, since surface temperature (and other meteorological variables), play an important role in not just ozone chemistry but in aerosol chemistry as well. More of thought than something that needs to be addressed in this article.

We thank the reviewer for bringing up these important points. Currently, we corrected cloud fields in WRF using satellite clouds only for radiation that is relevant to photochemistry. So, those cloud corrections do not affect other meteorological variables such as surface temperature, wind, boundary layer height, and so on. The cloud assimilation (cloud analysis more precisely) that has been used in the Rapid Refresh by NOAA (Benjamin et al., 2016), for example, uses satellite cloud products, and in the Rapid Refresh the thermodynamic balance between temperature and humidity due to the presence of clouds is considered, thus affecting temperature and humidity vertical profiles after cloud assimilation is applied. In addition, the addition or removal of clouds as a result of cloud assimilation affects surface radiation fields (such as downwelling solar radiation) and ultimately surface temperature and wind fields (and others). So, from the perspective of cloud data assimilation, the meteorological variables are all affected by cloud assimilation. We are planning to use the Rapid Refresh forecasts to conduct WRF-Chem simulations in the future and will report how cloud assimilation affects ozone forecasts.

The brief discussion regarding cloud assimilation is added in the last paragraph of conclusions and discussion section as follows.

*"The present study corrects cloud fields in WRF using satellite clouds only for radiation that is relevant to photochemistry, and those cloud corrections do not affect other meteorological variables such as surface temperature, wind, humidity, boundary layer height, etc. In a future study, we plan to examine the effects of satellite cloud assimilation on near-term $O_3$ forecasts using enhanced forecasts such as the Rapid Refresh products from NOAA (Benjamin et al., 2016) that take into account cloud data assimilation to derive meteorology for $O_3$ forecasts. The Rapid Refresh uses satellite cloud products as well as cloud observations from the ground and considers the thermodynamic balance between temperature and humidity due to the presence of clouds. Thus, this will allow investigating the effects of cloud assimilation on $O_3$ forecasts not only through changes in radiation for photochemistry but also through changes in meteorological variables."*

Specific comments:
C3) Line 14: What is meant by "attributed to that in cloud predictions"?
This was intended to mean "attributed to error in cloud prediction", and we revised it. So, the sentence is as follows.
*"It is not well known, however, how much error in $O_3$ predictions can be directly attributed to error in cloud predictions."*

C4) Line 45: Is surface ozone hourly? Perhaps specify if it is.
Yes. It is hourly ozone and we specified this in the revised manuscript.

C5) Line 206: Change "over CONUS" to "over the CONUS".
It is changed.

C6) Line 208: Remove "the" before central California.
It is removed.

C7) Line 211: Change "in supplementary" to "in the supplementary material".
It is changed.

C8) Line 288: I would be a little careful calling this 8-h average O3, since commonly 8-h average O3 refers to calculation of finding the maximum O3 across a number of 8-h averages throughout the day, whereas it appears the authors are simply using an afternoon average consisting of 8 hours. This might cause some confusion to some readers.

Following reviewers' comments, we have all replaced the daytime 8-h $O_3$ to maximum daily 8-h average (MDA8) $O_3$ in the revised manuscript.

C9) Line 361: This should be changed to say "partially corrected". It would be presumptions to assume that the cloud fields have been fully corrected. It is a big step in the right direction though.

We here intend to confine the correction to radiation fields relevant to photochemistry. So, the full sentence is revised as follows.

*"Still, large $O_3$ biases of ~11 ppb are present over the southeast US (compared to those of 6–9 ppb over CONUS) even though the clouds and radiation fields that are relevant to photochemistry are corrected."*

C10) Line 410: Remove "relatively" before greater.

It is removed.

C11) Fig 5. What is the cause of the very large reduction in O3 over the great lakes in the GOES simulation? Is that due to an improvement in clouds over the lakes themselves, or is it the result of improved clouds over the land and advection of O3 over the lakes? High O3 over the great lakes is a persistent problem in many air quality models, so the resulting improvement warrants some additional discussion in my opinion.

We thank the reviewer for bringing up this point. The correction of clouds both over the lakes and also in the upstream regions (mostly large cities located to the west/southwest of the lakes) can contribute to the reduction in $O_3$ bias. The precursors are emitted from the upstream cities and both the precursors and $O_3$ from the cities are advected toward the lakes where $O_3$ is not readily deposited over the water surface. We found that polluted air masses can be advected over the lakes. In this case in which precursor levels can be high over the lakes, the presence of clouds over the lakes can greatly affect $O_3$ formation over the lakes.

We added additional discussion in the revised manuscript:

*"The example shown here emphasizes the important roles of clouds in the Great Lakes region where large $O_3$ biases have been reported previously in air quality forecasts (e.g., Cleary et al., 2015). The correction of clouds both over the lakes and in the upstream regions (mostly large cities located to the west/southwest of the lakes) significantly reduces the $O_3$ bias. It is also shown that polluted air masses from the source regions can be advected over the lakes (not shown). In this case in which precursor levels can be high over the lakes, the presence of clouds over the lakes can greatly affect $O_3$ formation over the lakes."*

**Responses to Reviewer 3's comments**

Thank you so much for providing valuable comments. We have improved our manuscript following your suggestions and comments. Please find our responses below. Your comments are highlight in blue.

C1) The analysis is based primarily on one set of model physics (Morrison microphysics and Grell 3-D convection). The authors do test the sensitivity of the results to a second microphysics scheme (Thompson) and found little difference. However, the simulation is for summer conditions (June to September), when a significant amount of cloudiness is due to convection. Therefore, there should be a sensitivity test also run with a second convective scheme. I would suggest running the relatively new Grell-Freitas scheme. From what I have seen, this scheme will produce more clouds.

Following reviewer's suggestion, we have performed sensitivity tests with Grell-Freitas scheme. As done for microphysics scheme, a period of 10 days (3–12 July 2013) was considered. An example showing spatial distribution of cloud optical depth from the two cumulus parameterization schemes are presented in Fig. T1. In general, the spatial patterns and the location of large systems are similar to each other. The Grell-Freitas scheme produces more and/or thicker clouds in some regions such as the north Michigan and the south Ohio than the Grell-3D scheme. However, the Grell-Freitas scheme produces fewer and/or thinner clouds in other regions such as the east Texas and North Carolina. In Fig. T2, the histograms of cloud optical depth obtained for the 10-day period from Grell-Freitas scheme (left) and from Grell-3D scheme (right) show that the distributions of cloud optical depth are in general similar to each other. The Grell-Freitas scheme tends to produce fewer clouds with small or moderate cloud optical depth. Figure T3 shows that the degree of cloud correction in reducing $O_3$ bias is larger in VOC-limited regimes than in $NO_X$-limited regimes in the simulation with Grell-Freitas scheme, and thus the conclusions originally drawn remain unchanged.

We included the summary of this discussion above in the revised manuscript and figures (Figs. T2 and T3) in the supplementary materials.

[Figure]

Fig. T1. Cloud optical depth (COD) at 19 UTC 8 July 2013 using the (left) Grell-Freitas scheme and (right) Grell-3D scheme. Both simulations use the Morrison microphysics scheme.

[Figure]

Fig. T2. Histogram of hourly cloud optical depths during the daytime (16–23 UTC) over CONUS (land only) for the period of 3–12 July 2013 from simulations with the (left) Grell-Freitas scheme and (right) Grell-3D scheme.

[Figure]

Fig. T3. (Left column) The results of 3–12 July 2013 WRF-Chem simulations with Grell-Freitas scheme. (a/c) Probability density function of MDA8 $O_3$ bias (model value minus observation value) for VOC/$NO_X$-limited regime under cloudy sky conditions defined with COD threshold of 20 in the simulations with the Grell-Freitas scheme. (b/d) Same as (a/c), but for the simulations with the Grell-3D scheme. (e and f) Difference in median values of MDA8 $O_3$ bias between the two simulations with respect to COD threshold (i.e., CNTR minus GOES) for the simulations with the Grell-Freitas and with the Grell-3D schemes, respectively.

C2) In Section 2.3 the authors use the delta O3 to delta NOy ratio to determine VOC-limited and NOx-limited conditions. How is delta NOy determined at EPA monitoring sites? NOy is not routinely measured at these sites. Even true NOx is measured at only some small fraction of the O3 monitoring sites. This issue needs explanation or substantive revision.

$NO_y$ used in this study is the modeled $NO_y$ and $O_3$ is also modeled $O_3$. As you indicated, $NO_y$ is not routinely measured, so the sites having $NO_y$ measurements are very limited. Therefore, we could not rely on $NO_y$ observations. We included the following sentence in the revised manuscript.

"*Note that modeled $O_3$ and $NO_y$ in the CNTR simulation are used to determine whether an EPA site is in VOC-limited or $NO_X$-limited regime because $NO_y$ measurements are available for limited sites*."

In addition, we included examples showing how to determine VOC-limited or $NO_X$-limited sites in the supplementary materials (Fig. S1).

Minor comments:

C3) line 127: Which year NEI NOx was too high? Did Travis et al. indicate all NOx emission types were overestimated, or was it primarily mobile sources?

Travis et al. (2016) used 2011 NEI emissions and adjusted to 2013. They reduced NOx emissions from mobile and industrial sources (all sources except for power plants). Based on the references mentioned in Travis et al. (2016), several local studies reported that NEI NOx emissions for mobile sources are high by a factor of 2 or more (Castellanos et al, 2011; Fujita et al., 2012; Brioude et al., 2013; Anderson et al., 2014).

In our present study, we reduced $NO_X$ emission from all anthropogenic sources by 40% based on the analysis of Travis et al. (2016), and this is mentioned in the revised manuscript.

C4) lines 255 to 260: I don't follow this description of cloud fraction. Please clarify.
This part originally explained the results without showing figures that are relevant to the cloud fraction, but without showing figures we concluded that this part was too confusing to reader, and we decided to remove it. Please see the comment 12 of the first reviewer and our responses.

C5) Section 5.5 describes in detail how the box model calculations show that OH is less sensitive to changes in radiation in the NOx-limited regime. Some statements also need to be made about the effect on P(O3) in the box model.
Figure T4 shows the net chemical production of $O_3$ in the box model, and the result is consistent with that is found in the WRF-Chem simulations: larger sensitivity of P($O_3$) to cloudiness in VOC-limited regimes than NO$_X$-limited regimes. We briefly included this result in the revised manuscript as follows.
*"Note that the net chemical production of $O_3$ obtained from the box model results also shows a larger sensitivity to cloudiness in VOC-limited regimes than in NO$_X$-limited regimes (not shown)."*

[Figure]

Fig. T4. The net chemical production of $O_3$ from the box model simulations.

**Responses to the comments by Dr. P. Kasibhatla**

Thank you for providing valuable comments. Please find our responses (in black) to your comments (in blue) below.

This is an interesting paper that suggests a possible explanation for typical model over-prediction of surface ozone over CONUS (Figure 7). It is not clear however if model simulations are improved both in terms of surface $O_3$ predictions, as well as $O_3$ vertical profiles (especially in the boundary layer and just above the boundary layer). While comparisons with measured vertical profiles of $JNO_2$ are shown in Figure 3, no corresponding comparisons of vertical profiles are shown for $O_3$. It would be useful to show these comparisons (and provide histograms as is done for $JNO_2$) with simultaneous aircraft $O_3$ measurements, especially given the overprediction of $JNO_2$ in the boundary layer in the GOES simulation compared to the CNTR simulation for the NOMADSS flights (Figure 3).
In terms of model evaluation, it would be also useful to show comparisons of the modeled Ox vs NOz relationship against observations (as is done in Travis et al., 2016) as a check on modeled ozone production efficiency.

1. $O_3$ vertical profile comparison

The influence of satellite cloud corrections on vertical profile of $O_3$ is shown in Figs. P1 (SEAC$^4$RS) and P2 (NOMADSS). Only aircraft data over land within the southeast region (latitude: 25–40N, longitude: 95–70W) are used for the averages. Unlike the vertical profiles of $JNO_2$, which shows considerable improvements when satellite cloud corrections are applied, the vertical profiles of $O_3$ do not show significant differences between CNTR and GOES simulations even though the histograms of model-to-observation $O_3$ ratio show slight improvements in the GOES simulation than in the CNTR simulation. This is likely because the aircraft measurements are mostly made in rural environments or high altitudes where $O_3$ precursor concentrations are low. As shown in the manuscript, the effects of cloud correction are larger under high-$NO_X$ environments than low-$NO_X$ environments. An example on 21 September 2013 shows that GOES simulation better captures the attenuation of $JNO_2$ under below cloud conditions (~1830–1940 UTC) (Fig. P3). As the aircraft flew over relatively high-$NO_X$ regions during this time period, $O_3$ concentration shows a better agreement with observations in GOES simulation than CNTR simulation although both the simulations considerably overpredict $O_3$ in general. The largest difference in $O_3$ between the two simulations is 5.6 ppb at 1946 UTC. One of the reasons for the overprediction of $O_3$ could be the overprediction of $NO_2$ or misplacement of urban plumes. Other example for NOMADSS, on 7 July 2013 when the aircraft flew mostly over the state of Indiana and Lake Michigan shows similar results (Fig. P4). The sky conditions on that day were characterized by broken clouds, and the coarse resolution of satellite data (the original resolution is 8 km at hourly intervals) is another limitation for capturing the exact locations of small clouds. However, $O_3$ concentrations along the flight tracks in the two simulations show differences under cloudy conditions and the differences are noticeable only at high $NO_2$ (e.g., ~1635–1900 UTC). The largest difference in $O_3$ between the two simulations is 4.4 ppb at 1837 UTC. It should be noted that $O_3$ concentration in the two simulations is almost the same when $NO_2$ concentration is low even if $JNO_2$ values are significantly different (1920–1940 UTC). Thus, even though some cases show that clouds have significant influences on $O_3$ formation and concentrations above the ground (e.g., within the boundary layer), when all the data points are averaged the effects are hardly noticeable. We anticipate that if airborne measurements of $O_3$ under cloudy sky conditions are available over cities and/or urban plumes, then we would clearly see the effects of clouds on vertical profiles of $O_3$. Unfortunately, neither of the campaigns were designed for this purpose, so there are no good airborne observation data to examine the effects of clouds on vertical profiles of $O_3$.

[Figure]

Fig. P1. (Top) (Left) Cloudy-sky averaged vertical profiles of $O_3$ for SEAC$^4$RS observations, CNTR and GOES simulations. (Middle and Right) Histogram of ratio of $O_3$ simulated by the model to $O_3$ observed for CNTR simulation and GOES simulation, respectively.

[Figure]

Fig. P2. Same as Fig. P1, but for NOMADSS campaign.

[Figure]

Fig. P3. An example for SEAC⁴RS campaign (21 September 2013). (Top, left) Timeseries of aircraft altitude. Shading indicates cloud boundaries from GOES retrievals. (Top, right) Timeseries of NO₂ concentration. (Bottom, left) Timeseries of JNO₂. (Bottom, right) Timeseries of O₃ concentration. Note that the shorter time period than the whole flight-day time period is shown here to highlight the effects of clouds.

[Figure]

Fig. P4. Same as Fig. P3, but for a NOMADSS example (7 July 2013).

2. Ozone production efficiency evaluation

The ozone production efficiency (OPE) is evaluated against SEAC[4]RS observations over the southeast US (Fig. P5). The OPE from the model (14.3) is similar to that from the observations (14.0), showing a good performance of our model. Both OPE values are smaller than the values shown in Travis et al. (2016); 16.7 for their model and 17.4 for SEAC[4]RS observations. Even though we use the same criteria as in Travis et al. (2016) such as altitudes lower than 1.5 km and $NO_Z = HNO_3 + PAN + $ aerosol nitrate + alkyl nitrates, we do not exclude urban plumes and open fire plumes because 1) we are interested in urban areas and urban plumes and 2) the condition of filtering out open fire plumes (using $CH_3CN$) may not be appropriate to apply to our relatively high resolution simulations. When urban plumes and open file plumes are excluded in the SEAC[4]RS observations (not shown), we find a very similar value of observed OPE (17.46) as in Travis et al.'s value (17.4).

[Figure]

Fig. P5. Ozone production efficiency (OPE) below 1.5 km over the southeast US for SEAC[4]RS campaign. $O_X$ is $O_3 + NO_2$, and $NO_Z$ is $HNO_3$ + PAN + aerosol nitrate + alkyl nitrates.

[revised manuscript text omitted]